# Tracking Functional Changes in Nonstationary Signals with Evolutionary Ensemble Bayesian Model for Robust Neural Decoding

**Xinyun Zhu**[1,2,4]**, Yu Qi**[3,4,*]**, Gang Pan**[4,2]**, Yueming Wang**[1,4,*]

{zhuxinyun, qiyu, gpan, ymingwang}@zju.edu.cn

[1]Qiushi Academy for Advanced Studies, Zhejiang University, Hangzhou, China

[2]College of Computer Science and Technology, Zhejiang University, Hangzhou, China

[3]Affiliated Mental Health Center & Hangzhou Seventh People's Hospital and the MOE Frontier Science Center for Brain Science and Brain-machine Integration, Zhejiang University School of Medicine, Hangzhou, China.

[4]The State Key Lab of Brain-Machine Intelligence, Zhejiang University, Hangzhou, China

## Abstract

Neural signals are typical nonstationary data where the functional mapping between neural activities and the intentions (such as the velocity of movements) can occasionally change. Existing studies mostly use a fixed neural decoder, thus suffering from an unstable performance given neural functional changes. We propose a novel evolutionary ensemble framework (EvoEnsemble) to dynamically cope with changes in neural signals by evolving the decoder model accordingly. EvoEnsemble integrates evolutionary computation algorithms in a Bayesian framework where the fitness of models can be sequentially computed with their likelihoods according to the incoming data at each time slot, which enables online tracking of time-varying functions. Two strategies of evolve-at-changes and history-model-archive are designed to further improve efficiency and stability. Experiments with simulations and neural signals demonstrate that EvoEnsemble can track the changes in functions effectively thus improving the accuracy and robustness of neural decoding. The improvement is most significant in neural signals with functional changes.

## 1 Introduction

Nonstationary signals widely exist in the real world, where the properties and functions can change continuously over time. Neural signals are typical nonstationary signals, where the inherent dynamics in the neural systems, the plasticity of synapses driven by learning and adaptation contribute to the variability of neural encoding of the brain [1, 2]. The nonstationary property of neural signals forms a challenging problem in the brain-computer interfaces (BCIs) field [3–9], because the decoding accuracy will degrade over time given changing neural functions and a fixed neural decoder.

Considering the nonstationary of neural signals, the neural encoding model, can be presented by:

$$\boldsymbol{y}_t = h_t(\boldsymbol{x}_t) + \boldsymbol{q}_t, \tag{1}$$

where $\boldsymbol{y}_t$ denotes the neural signal we observed at time $t$, the function $h_t(\cdot)$ is the neural encoding model that changes over time, and $\boldsymbol{x}_t$ denotes the state to be encoded such as the velocity of a computer cursor, and $\boldsymbol{q}_t$ is biological or external noises. Typical neural decoders such as OLE [10] and Kalman filters (KF) [11] mostly assume a stationary $h_t(\cdot)$, namely $h_t(\cdot) \equiv h_0(\cdot), (t = 1, 2, 3, ...)$,

---

*Corresponding authors: Yu Qi and Yueming Wang

36th Conference on Neural Information Processing Systems (NeurIPS 2022).

which can be oversimplified especially for online BCI systems. Recent studies showed that, in online BCI control, neural encoding functions change significantly with the influence of real-time feedback such as the speed of the cursor and errors in control [12, 13]. The functional changes in neural encoding frequently occur over time, even in a single target reaching trial [14].

How to cope with the functional changes in the neural system? Ideally, a neural decoder should be capable of adjusting itself along with changes in neural signals. To this end, there are two main types of solutions. The first one is re-calibration of neural decoders periodically or manually when the performance degrades [15, 16] to maintain the control accuracy. Brandman *et al.* proposed a framework for rapid calibration, which removed the traditional open-loop calibration phase and reduced the original 2-5 min calibration interval to 2-5 s with Bayesian parameter updating. However, such a training process can still disturb the BCI users and usually can not cope with changes in the short-term such as single trials. Another way is to design an adaptive or dynamic model [17–19]. Wang *et al.* proposed a framework that used dual state-space models to estimate both kinematics and neural tuning functions. However, this method does not directly model the neural variability. Qi *et al.* proposed a novel dynamic ensemble Bayesian filter (DyEnsemble) which obtained the state-of-the-art performance with nonstationary neural signals [20]. DyEnsemble constructs a pool of encoding models, and adaptively assembles a decoder from these models online according to neural signals with a recursive Bayesian filter, which allows the neural decoder adjust its functions to cope with the changes in neural signals. However, DyEnsemble employs a static model pool, only addressing neural changes in a certain range. The ability can be limited given continuous functional changes in long-term neural recordings.

Regarding the encoding changes in neural signals as a functional tracking problem, we propose an evolutionary ensemble Bayesian filter (EvoEnsemble) to track the neural encoding functions over time by incorporating evolutionary computation in a recursive Bayesian framework. Specifically, EvoEnsemble maintains a population (i.e., a set of candidate functions) to estimate the neural functions, where the changing fitness of the population is dynamically computed by the likelihood given neural signals (observation) over time. In this way, the population evolution can be driven by the changes in neural signals. Meanwhile, the time-varying functions can be tracked by the posterior distribution of the candidate functions with Bayesian model averaging rules.

The main contributions of this study can be summarized as follows: (1) We propose a new framework that extends evolutionary computation in a Bayesian process to achieve robust state estimation with neural functional changes; (2) We propose a particle-based solution to simultaneously track the functional changes and estimate the state (e.g., the velocity of a cursor) sequentially online; (3) Two strategies of evolve-at-changes and history-model-archive are proposed to improve the computational efficiency and estimation stability.

Experiments with simulations and neural signals demonstrate the superiority of EvoEnsemble against traditional decoders such as Kalman filters, and the improvement is most significant with functional changes in neural signals.

## 2   State-space formulation with time-varying neural functions

Considering the temporal changes in neural functions, the state-space model can be defined by

$$\boldsymbol{x}_t = g(\boldsymbol{x}_{t-1}) + \boldsymbol{n}_{t-1}, \tag{2}$$

$$\boldsymbol{y}_t = h_t(\boldsymbol{x}_t) + \boldsymbol{q}_t, \tag{3}$$

where $t$ denotes the time slot. $\boldsymbol{x}_t \in \mathbb{R}^{d_z}$ is the state we want to estimate; $\boldsymbol{y}_t \in \mathbb{R}^{d_y}$ is the observation; $g(\cdot)$ is the state transition function, with a Gaussian transition noise $\boldsymbol{n}_t$; and $h_t(\cdot)$ is the time-varying observation function which is dependent on $t$, with a Gaussian observation noise $\boldsymbol{q}_t$.

In a typical motor decoding problem, given a sequence of observed neural signals, a neural decoder aims to estimate the corresponding kinematic states in a sequential manner. Here we define the state $\boldsymbol{x}_t$ with $d_z = 2$, which contains the velocities of movements in x-axis and y-axis, and $\boldsymbol{y}_t$ is the firing rates of $d_y$ neurons.

The EvoEnsemble approach consists of two phases, one is the calibration phase and the other is the test phase. During the calibration phase, training sample pairs are collected and used to learn the transition function, as well as a set of initial encoding models to approximate the time-varying

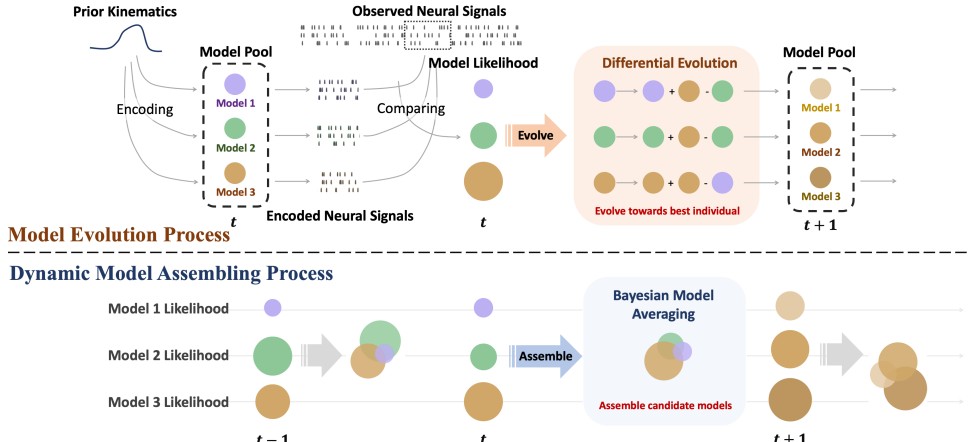

Figure 1: Illustration of the model evolution process (above) and the dynamic model assembling process (below). At each time slot, EvoEnsemble encodes the prior kinematics with candidate models and then calculates the model likelihoods through comparing the encoded neural signals with the observed neural signals. Afterwards, the model likelihoods are used to evolve the model pool and assemble the models simultaneously.

observation function. Once the calibration phase is over, the transition function is fixed. Meanwhile, a set of initial particles are randomly drown from a Gaussian distribution, to approximate the posterior of the kinematic states. During the test phase, the particle likelihoods and model likelihoods are calculated through comparing the neural signals encoded by the models with the observed ones. The observation function changes along with the incoming neural signals through adaptively assembling the encoding models in the model pool, where the assembling weights (i.e., the posterior probabilities of the models) are updated from the model likelihood. When all models' likelihoods in the model pool are small, the model pool will be updated by EvoEnsemble through a differential evolution algorithm, where model fitness is equal to the model likelihood.

## 3    Tracking functional changes with EvoEnsemble

EvoEnsemble tracks the changes of function $h_t(\cdot)$ in Eq. (3) with a population of candidate functions (model pool) which evolves with changes in data over time. Then it estimates $h_t(\cdot)$ at each time slot $t$ by assembling the candidate functions in the model pool with a Bayesian model averaging algorithm. To this end, there are several key problems: 1) how to evolve the model pool over time; 2) how to dynamically assemble the model pool to an optimized estimation of $h_t(\cdot)$; and 3) how to sequentially estimate the state $\boldsymbol{x}_t$ given $h_t(\cdot)$.

Focusing on the problems above, the EvoEnsemble can be divided into three main components: 1) a population evolution algorithm to evolve the model pool in time (See Fig. 1 above); 2) a dynamic Bayesian model ensemble approach estimate $h_t(\cdot)$ from the model pool (See Fig. 1 below); and 3) a particle filter to estimate $\boldsymbol{x}_t$ given $h_t(\cdot)$. Further, two strategies of evolve-at-changes and history-model-archive are proposed to improve the efficiency and stability of the sequential process.

### 3.1    Population evolution in the Bayesian framework

In this section, we first introduce how to initialize the population, and then show how to dynamically compute the fitness of the population with the model likelihoods in the Bayesian framework. After that, we present the evolutionary process with the differential evolution (DE) algorithm.

#### 3.1.1    Population initialization

The candidate model pool, $\mathcal{M} = \{m^k(\cdot)|k = 1, 2, ..., N\}$, is first initialized by a set of encoding models that transform kinematics into neural signals. To enrich the diversity of the model pool, each model is trained with a different subset of the training data, which contains a randomly divided segment. Here we use linear encoding models and the models are fitted with the least square algorithm.

We denote the linear mapping matrix as $\mathbf{M}^k$, and its parameters as a vector $\mathbf{p}^k$. Thus, for linear case, the candidate model pool can also be denoted as $\mathcal{M} = \{\mathbf{p}^k | k = 1, 2, ..., N\}$, where $\mathbf{p}^k$ is the $k^{th}$ individual in the evolution population. The specific operation of getting subset indices is given in Appendix A.

### 3.1.2 Fitness definition by the model likelihood

The purpose of evolution is to search for the best individual in the population and evolve other individuals towards it. Therefore, a fitness function that evaluates the quality of the model plays a core role in evolutionary computation.

In the Bayesian framework, the fitness of a model can be defined by its likelihood given observation data. A model with a high likelihood demonstrates good fitness. Since the candidate models in the model pool are neural encoding functions, which predict neural signals given kinematic parameters, we firstly infer the prior distribution of kinematics, namely $\boldsymbol{x}_t$, by Eq. (2), then for each candidate function $m^k(\cdot)$ we can infer the neural signal $\hat{\boldsymbol{y}}_t^k$ by Eq. (3), and compute the likelihood $l^k$ of the observed neural signal $\boldsymbol{y}_t^k$ given $\hat{\boldsymbol{y}}_t^k$. The likelihood $l^k$ indicates the fitness of $m^k(\cdot)$. The fitness computation process is specified as follows.

**Step1: Prior kinematics prediction.** First, given the kinematics $\boldsymbol{x}_{t-1}$ at time $t-1$, we can get the prior kinematics $p(\boldsymbol{x}_t|\boldsymbol{x}_{t-1})$ at time $t$ through the state transition function $g(\cdot)$ in Eq. (2).

**Step2: Neural encoding with different models.** Second, we can generate different neural signals from the prior kinematics with different encoding models in the model pool. The neural signals encoded by the $k^{th}$ model at time $t$ is $\hat{\boldsymbol{y}}_t^k = m^k(\boldsymbol{x}_t) = \mathbf{M}^k \boldsymbol{x}_t$.

**Step3: Likelihood computing for each model.** Third, we compute each model's likelihood by comparing its encoded neural signals $\hat{\boldsymbol{y}}_t^k$ with the incoming neural signals $\boldsymbol{y}_t$. One model's likelihood indicates how likely the observed neural signals are encoded by this model. The more likely it is, the greater the value is. The marginal likelihood of the $k^{th}$ model, given the observed neural signals at time $t$ can be represented as follows:

$$p^k(\boldsymbol{y}_t|\boldsymbol{y}_{0:t-1}) = \int p^k(\boldsymbol{y}_t|\boldsymbol{x}_t)p(\boldsymbol{x}_t|\boldsymbol{y}_{0:t-1})d\boldsymbol{x}_t, \qquad (4)$$

where the $p(\boldsymbol{x}_t|\boldsymbol{y}_{0:t-1})$ is the probability of a certain kinematic given all the neural signals before, and $p^k(\boldsymbol{y}_t|\boldsymbol{x}_t)$ is the model likelihood given the certain kinematic. A particle-based solution towards it will be given in Section 3.3, where the model likelihoods are obtained by summing the likelihoods of a set of particles.

**Step4: Fitness definition by the model likelihood.** Finally, we define the fitness value of the individual $\mathbf{p}^k$ at time $t$ using the likelihoods of recent time slots:

$$f_t(\mathbf{p}^k) = \frac{1}{l_{\text{pre}}} \sum_{j=t-l_{\text{pre}}}^{t} p^k(\boldsymbol{y}_j|\boldsymbol{y}_{0:j-1}), \qquad (5)$$

where $l_{\text{pre}}$ is the length of the recent time slots. Note that, the neural decoding is a sequential problem, and we do not update the model pool at each time point. That is, we usually accumulate data for several previous time slots for one update process.

### 3.1.3 Population evolution with the DE

With the initial model pool and the fitness function, we can evolve the population with evolutionary computation algorithms. We use the DE algorithm with real-value encoding rather than those evolutionary computation algorithms with binary encoding. DE is a global optimization algorithm that follows an iterative population initialization, mutation, crossover and selection procedure. The canonical DE is introduced in Appendix B, and here we present the DE in model evolution.

**Current-to-pbest mutation.** Mutation obtains evolution directions from individual differences. A mutated population of the $G^{th}$ generation $\{\mathbf{v}_G^k | k = 1, 2, ..., N\}$ comes from the parent population $\{\mathbf{p}_G^k | k = 1, 2, ..., N\}$ according to a certain mutation strategy.

To avoid the premature convergence problem and diverse the population [21], we adopt a "DE/current-to-pbest/1" with archive strategy following [22]:

$$\mathbf{v}_G^k = \mathbf{p}_G^k + F_k \cdot (\mathbf{p}_G^{\text{pbest}} - \mathbf{p}_G^k) + F_k \cdot (\mathbf{p}_G^{r_k^1} - \tilde{\mathbf{p}}_G^{r_k^2}), \tag{6}$$

where $\mathbf{p}_G^{\text{pbest}}$ is randomly chosen from the top $100 * p\%$ of the population, and usually $p \in [0.05, 0.2]$. $r_k^1$ and $r_k^2$ are randomly selected indices in the range of $[1, N]$, being distinct from each other and also the index $k$. The $F$ is a list of the positive mutation factor and $\tilde{\mathbf{p}}_G^{r_k^2}$ is randomly selected from a collection storing the inferior individuals. See more details in Appendix C.

**Crossover.** Crossover randomly selects component from either parent or mutated vectors in each dimension. The individual after crossover is a D-dimension vector $\{\boldsymbol{u}_G^k = \boldsymbol{u}_{1,G}^k, \boldsymbol{u}_{2,G}^k, ..., \boldsymbol{u}_{D,G}^k\}$:

$$\boldsymbol{u}_{j,G}^k = \begin{cases} \boldsymbol{v}_{j,G}^k, & \text{if } \text{rand}(0,1) \leq CR_k \text{ or } j = j_{\text{rand}} \\ \boldsymbol{p}_{j,G}^k, & \text{otherwise} \end{cases}, \tag{7}$$

where $j_{\text{rand}}$ is randomly chosen from $[1, D]$, and the $CR$ is a list of the positive crossover factor.

**Selection.** In the selection stage, the fitness function $f(\cdot)$ determines whether an individual stays in the new population or not:

$$\mathbf{p}_{G+1}^k = \begin{cases} \mathbf{u}_G^k, & \text{if } f(\mathbf{u}_G^k) > f(\mathbf{p}_G^k) \\ \mathbf{p}_G^k, & \text{otherwise} \end{cases}. \tag{8}$$

Usually, one model pool evolution process iterates for hundreds of generations (mutation, crossover, selection) until it satisfies the pre-set conditions.

## 3.2 Estimating $h_t(\cdot)$ with dynamic Bayesian model ensemble

A Bayesian model ensemble approach [20] is employed to estimate $h_t(\cdot)$ at each time slot given the model pool. Specifically, $h_t(\cdot)$, can be represented as a weighted combination of models in the pool by:

$$h_t(\cdot) = \sum_{k=1}^{N} w_t^k \cdot m^k(\cdot), \tag{9}$$

where $w_t^k$ is the assembling weight, changing with neural signals over time. It can be dynamically obtained with the dynamic Bayesian ensemble algorithm.

Specifically, the assembling weight $w_t^k$ of each model $m^k(\cdot)$ at time $t$ is computed by the posterior probability of each model given the observation neural signals $\boldsymbol{y}_t$

$$p(h_t(\cdot) = m^k(\cdot)|\boldsymbol{y}_{0:t}) = \frac{p(h_t(\cdot) = m^k(\cdot)|\boldsymbol{y}_{0:t-1})p^k(\boldsymbol{y}_t|\boldsymbol{y}_{0:t-1})}{\Sigma_{j=1}^{N} p(h_t(\cdot) = m^j(\cdot)|\boldsymbol{y}_{0:t-1})p^j(\boldsymbol{y}_t|\boldsymbol{y}_{0:t-1})}, \tag{10}$$

where $p(h_t(\cdot) = m^k(\cdot)|\boldsymbol{y}_{0:t-1})$ is the prior probability of $m^k(\cdot)$ at time $t$, which is given by the posterior probability of $m^k(\cdot)$ at time $t-1$; and $p_k(\boldsymbol{y}_t|\boldsymbol{y}_{0:t-1})$ is the likelihood as in Eq. (4).

## 3.3 Estimating $\boldsymbol{x}_t$ with a particle-based solution

With $h_t(\cdot)$ at hand, we can estimate the posterior distribution of state $\boldsymbol{x}_t$ by

$$p(\boldsymbol{x}_t|\boldsymbol{y}_{0:t}) = \sum_{k=1}^{N} p(\boldsymbol{x}_t|h_t(\cdot) = m^k(\cdot), \boldsymbol{y}_{0:t})p(h_t(\cdot) = m^k(\cdot)|\boldsymbol{y}_{0:t}), \tag{11}$$

where $p(\boldsymbol{x}_t|h_t(\cdot) = m^k(\cdot), \boldsymbol{y}_{0:t})$ is the posterior probability of the state with association to the $k^{th}$ model, which can be estimated recursively with the Bayesian filter (i.e., the particle filter). And $p(h_t(\cdot) = m^k(\cdot)|\boldsymbol{y}_{0:t})$ is the model weight coming from Eq. (10) with Bayesian model averaging rules.

Specifically, we propose a particle-based solution to compute the model likelihoods and state posterior simultaneously (see Appendix D for details). Suppose there is a set of particles $\{\boldsymbol{x}_t^{(1)}, \boldsymbol{x}_t^{(2)}, ..., \boldsymbol{x}_t^{(N_{\text{par}})}\}$, and the normalised importance weights of particle $s$ under the hypothesis $h_t(\cdot) = m^k(\cdot)$ are given by $\boldsymbol{\omega}_t^{k(s)}, s = 1, 2, ..., N_{\text{par}}$, the particle-based marginal likelihood can

be calculated as follows:

$$p^k(\boldsymbol{y}_t|\boldsymbol{y}_{0:t-1}) = \sum_{s=1}^{N_{\text{par}}} \boldsymbol{\omega}_{t-1}^{k(s)} p^k(\boldsymbol{y}_t|\boldsymbol{x}_t^{(s)}), \tag{12}$$

where $\sum_{s=1}^{N_{\text{par}}} \boldsymbol{\omega}_t^{k(s)} = 1$. Further, the particle-based state posterior can be calculated by:

$$p(\boldsymbol{x}_t|h_t(\cdot) = m^k(\cdot), \boldsymbol{y}_{0:t}) \approx \sum_{s=1}^{N_{\text{par}}} \boldsymbol{\omega}_t^{k(s)} \boldsymbol{\delta}(\boldsymbol{x}_t - \boldsymbol{x}_t^{(s)}), \tag{13}$$

where $\boldsymbol{\delta}$ is the Dirac delta function. When the $N_{\text{par}} \to \infty$, the approximation approaches the true posterior density.

### 3.4  Strategies for model pool update

We propose two strategies, namely evolve-at-changes and history-model-archive, to decide when to update and how to update the model pool, respectively.

**The evolve-at-changes strategy.** The default setting of the EvoEnsemble is to update the model pool at a regular interval $t_{\text{up}}$. However, too large an interval may lead to untimely updates, while too small an interval may lead to unnecessary updates. Evolve-at-changes is a strategy to choose the update time adaptively according to the models' likelihoods. Specifically, we maintain a list of log-likelihood $L_{\text{max-ever}}$ where keeps the maximum log-likelihoods of all the candidate models in each time slot. Every time a new neural signal comes in, we average the latest three values in $L_{\text{max-ever}}$ as current log-likelihood $l_{\text{cur}}$, and average the three values before the latest three values in $L_{\text{max-ever}}$ as previous log-likelihood $l_{\text{pre}}$. Meanwhile, we add a ratio $r_{\text{up}}$ for flexibility. Thus, the model pool will update once $l_{\text{cur}} < r_{\text{up}} \cdot l_{\text{pre}}$. Besides, we limit the interval between two updates to 3 or more.

**The history-model-archive strategy.** The default setting of the EvoEnsemble is to update the model pool using all individuals evolved by DE. However, DE tends to evolve the models all towards the best model and leaves the model pool lacking diversity, which is insufficient to characterize the variability of neural signals. Thus, we establish a history model archive $\mathbf{m}_{\text{his}}$, to store the best models in each time slot, whose size is in line with the main model pool. When the iterations of DE finish, we randomly choose models in the history model archive, replacing the models in the evolved model pool. Again, we define a ratio $r_{\text{pre}}$ to determine how many history models to be preserved in the main model pool, which can be written as $N_{\text{pre}} = r_{\text{pre}} \cdot N$. So the final updated model pool is $\{\mathbf{p}^k|k = 1, 2, ..., N\} = \{\mathbf{p}_{\text{his}}^{r_1}, \mathbf{p}_{\text{his}}^{r_2}, ..., \mathbf{p}_{\text{his}}^{r_{N_{\text{pre}}}}, \mathbf{p}^{N_{\text{pre}}+1}, \mathbf{p}^{N_{\text{pre}}+2}, ..., \mathbf{p}^N\}$.

## 4  Experiments and results

### 4.1  Performance with simulations

To evaluate the performance of EvoEnsemble with different conditions of functional changes, we simulate several conditions of changing functions, including functions with slow or rapid monotonic changes, functions with slow or rapid non-monotonic changes, and a piece-wisely changing function (as shown by the red lines in Fig. 2 (a), see details in Appendix D). Specifically, the function's first parameter for conditions 1 - 4 is the same: $h_t^1 = 7\text{E-}3t + 1, \quad 1 \le t \le 300$, while the second parameters are:

(1) $h_t^2 = 7\text{E-}4t - 1.9$;

(2) $h_t^2 = 1.12\text{E-}2t + 2.6$;

(3) $h_t^2 = 9.8\text{E-}6t^2 - 2.8\text{E-}3t + 3.4$;

(4) $h_t^2 = -3.43\text{E-}5t^2 + 4.2\text{E-}3t - 1.7$;

For condition 5, the parameters piece-wisely change over time:

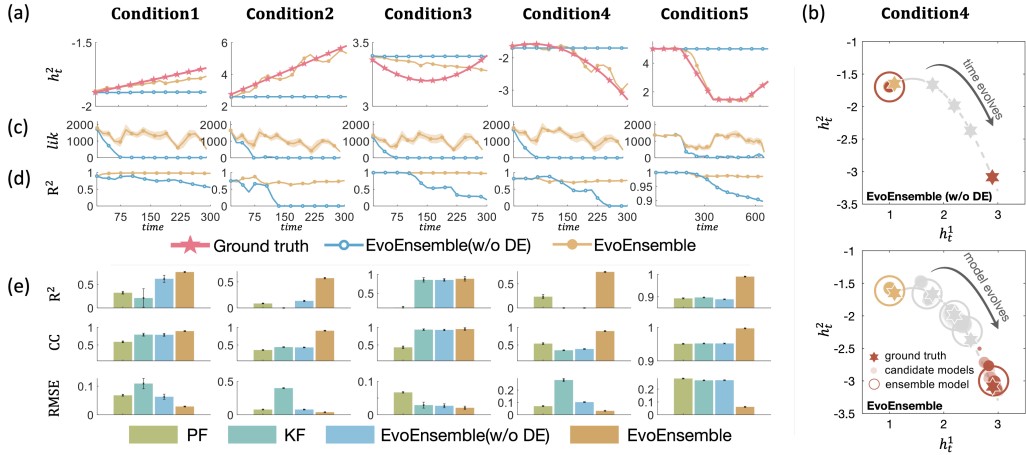

Figure 2: (a) Function parameter $h_t^2$ tracking performance of EvoEnsemble (yellow) and EvoEnsemble(w/o DE) (blue) in five conditions, comparing with the ground truth (pink). (b) Visualization of model ensemble and model evolution process with EvoEnsemble(w/o DE) (above) and EvoEnsemble (below). (c) Comparisons of candidate model likelihoods change. (d) Comparisons of decoding performance change. (e) Overall performance of $R^2$, CC, RMSE in five conditions comparing between four different decoders.

$$(5) \quad h_t^1 = \begin{cases} 4, & 0 < t \le 168, \\ 1E\text{-}2t + 2.32, & 168 < t \le 345, \\ 5.77, & 345 < t \le 517, \\ -2E\text{-}2t + 1.611E1, & 517 < t \le 650; \end{cases} \quad h_t^2 = \begin{cases} 5, & 0 < t \le 168, \\ -2E\text{-}2t + 8.36, & 168 < t \le 345, \\ 1.46, & 345 < t \le 517, \\ 1E\text{-}2t - 3.71, & 517 < t \le 650. \end{cases}$$

**Performance of functional tracking.** We first evaluate the functional tracking ability of EvoEnsemble with different conditions. Fig. 2 (a) illustrates the ground truth (red lines) and the EvoEnsemble estimated (yellow lines) function's parameters $h_t^2$ compared with EvoEnsemble without DE-based model evolution (blue lines). Results demonstrate that EvoEnsemble tracks the changes in functions closely over time for all five conditions. To further analyze how EvoEnsemble tracks the functional changes with the evolutionary model pool, we present the internal tracking process within EvoEnsemble in Fig. 2 (b). It shows that the candidate functions in the model pool can effectively follow changes of the true function $h_t$ with the model evolution process, such that the dynamic estimation of $h_t$ is obtained. To quantitively evaluate the functional tracking process, we compute the likelihood of the functions estimated over time in Fig. 2 (c). EvoEnsemble keeps a high likelihood to the neural data with different changes, suggesting that EvoEnsemble can effectively capture the changing functions in time.

**Performance of state estimation.** Here we evaluate the state estimation performance with different criteria of correlation coefficient (CC), root mean squared error (RMSE), and determination of coefficients ($R^2$) between the estimated and ground-truth state sequences. Fig. 2 (d) illustrates the $R^2$ of the estimated states. Results show that, without the model evolution process, the performance decreases gradually as the function changes, and the $R^2$ decrease to 0.552, 0, 0.187, 0, and 0.896 with the five conditions respectively. EvoEnsemble is more robust to functional changes that the $R^2$s are 0.975, 0.759, 0.970, 0.764, and 0.986 for the five conditions, respectively. Fig. 2 (e) further compare the state estimation performance with different neural decoders. Overall, EvoEnsemble achieves the superior performance with all the conditions, with stable high $R^2$, CC and low RMSE. Specifically, EvoEnsemble obtains CCs of 0.894, 0.905, 0.952, 0.895, and 0.997 with the five conditions, which are 12.6%, 105.2%, 1.7%, 155.0%, and 4.7% higher than Kalman filters (see details in Appendix E).

These results suggest the necessity and importance of functional tracking for robust state estimation with time-varying functions, and demonstrate that EvoEnsemble can track the changing functions effectively to improve the accuracy and robustness of state estimation.

**Effectiveness of the evolve-at-changes strategy.** The evolve-at-changes strategy aims to control 'when to update' the model pool to balance the trade-off between efficiency and accuracy: frequently

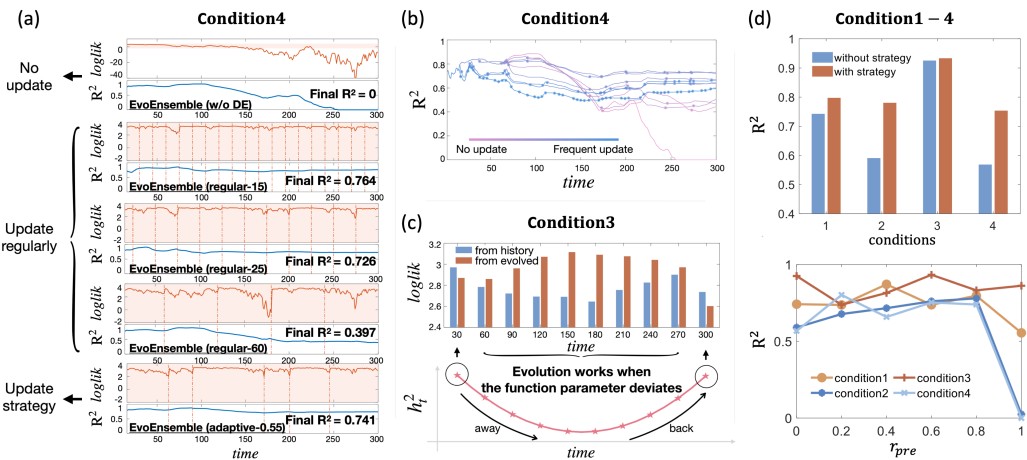

Figure 3: (a) The changes of log-likelihood and decoding $R^2$ in five different model update settings, where the dashed line represents the update time point. (b) The changes of $R^2$ over time under different update ratio $r_{up}$ settings, where the asterisks denote the update time point. (c) The maximum log-likelihood of the models coming from history and from evolution after each update (above) in condition 3 (below). (d) Comparison of the decoding $R^2$ between EvoEnsemble without history-model-archive strategy and with history-model-archive strategies (above). The decoding $R^2$ under different historical model preserving ratios $r_{pre}$ (below).

updating the model pool helps track the changes in functions more precisely while requiring more computational costs. Here we compare the state estimation performance using different model updating rules with simulation 4. Fig. 3 (a) compares the log-likelihood of models and the state estimation $R^2$ with five different settings of 1) no updating; 2) updating with a regular time interval; and 3) updating with the evolve-at-changes strategy. Results demonstrate that with the proposed strategy, EvoEnsemble achieves a high $R^2$ of 0.741 comparable to 0.764, while reducing 72.2% of model update costs, which well balances the efficiency and accuracy. Fig. 3 (b) analyzes the influence of updating ratio $r_{up}$ using $R^2$ together with the update timing (with the asterisks). Results demonstrate that the decoding performance is relatively stable when the update ratio is 0.4 to 0.9.

**Effectiveness of the history-model-archive strategy.** History-model-archive is proposed to improve the stability of state estimation with the information of historical models. Fig. 3 (c) shows the maximum log-likelihoods of the models from the historical model pool (blue bars) and the current model pool (red bars) after every update in condition 3, where a non-monotonic functional change is involved (as shown in the lower panel of Fig. 3 (c)). From time slots 0 to 150 where the function gradually deviates from the initial, models in the current model pool gradually take the dominant part to track the changing functions with model evolution. While from time slots 180 to 300 where the function comes back to the initial, the likelihood of historical models gradually increases. We further evaluate the decoding performance of $R^2$ with different history preserve ratio $r_{pre}$. Fig. 3 (d) (above) shows that the history-model-archive strategy improves the decoding performance in all five conditions. Fig. 3 (d) (below) tests the influence of history preserve ratio ($r_{pre}$) parameter in the history-model-archive strategy, which shows that the optimal $r_{pre}$ relies on the conditions of data. Thus $r_{pre}$ can be selected as a hyper parameter for each dataset.

### 4.2 Performance with neural signals

We evaluate our EvoEnsemble approach on four datasets with neural signals recorded from a macaque monkey (Data-M) and a human with tetraplegia (Data-P1/2/3).

**Monkey neural datasets.** The monkey data is from the open Zenodo dataset [23] where the monkeys were trained to perform a self-paced reaching task in an $8 * 8$ grid (as shown in Fig. 4 (a)). Neural spike signals are recorded from the M1 area. We use the data segment of indy-2017012401, and we select 20 channels according to the CC between neural signals and kinematics. The neural data are smoothed with a window of 3 time slots. The hand velocities are obtained from the position by a discrete derivative.

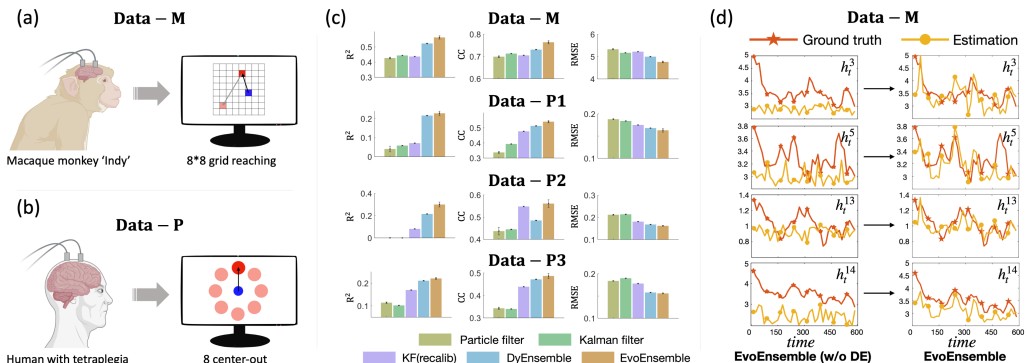

Figure 4: (a) Illustration of the grid reaching paradigm with a macaque monkey. (b) Illustration of the center-out paradigm with a human participant. (c) The neural decoding performance $R^2$, CC, RMSE of the four neural datasets. (d) The tracking performance for four exemplary parameters of Data-M with EvoEnsemble(w/o DE) and EvoEnsemble.

**Clinical neural datasets.** The clinical dataset was collected with a human subject, with two 96-channel Utah arrays implanted in the left primary motor cortex. All the clinical and experimental procedures were approved by the Medical Ethics Committee. The BCI experiment paradigm is a center-out 2D cursor control with eight directions (as shown in Fig. 4 (b)). In the online closed-loop BCI control, there is a calibration phase with two observation blocks and three ortho-impedance assistant training blocks similar to [24], where the ortho-impedance assistance ratios of the three blocks are 0.7,0.5, and 0.3 respectively. After the training blocks, a full brain control block where no ortho-impedance assistance is applied. We use the training blocks for decoder training, and the sixth block for the test. Neural signals are sorted manually before each day's experiment. Similar to Data-M, 20 neurons with top CCs between neural signals and kinematics are selected. Data-P1/2/3 were collected in 20201208/09/14 respectively.

**Neural decoding performance.** The neural decoding performance with Data-M and Data-P1/2/3 is presented in Table 1 and Fig. 4 (c). Overall, EvoEnsemble outperforms the other decoders in all the datasets. Compared with the static decoders such as the Kalman filter, EvoEnsemble obtains CCs of 0.764, 0.547, 0.561 and 0.487 with Data-M and Data-P1/2/3 respectively, which are 7.2%, 39.2%, 26.1% and 43.2% higher than the Kalman filter. The improvement is most significant in the Data-P datasets with closed-loop BCI control. We think it is partly because that the feedback in the closed-loop BCI control process may lead to frequent functional changes in neural signals (as shown in Fig. S1). The functional changes lead to unstable performance with static approaches of particle filters (PF) and Kalman filters, such that low CCs of 0.337, 0.437, 0.343 (PF) and 0.393, 0.445, 0.340 (KF) are obtained. With the DyEnsemble approach [20] where the neural decoder dynamically adjusts along with changes in neural signals, the performance increases to 0.513, 0.485, and 0.472 for the three datasets respectively. However, without the model evolution process, the performance is limited given substantial changes in functions (in Data-P1/2, see Fig. S1). While EvoEnsemble further improves the performance by 6.6% and 15.7% on Data P1/2 with the functional tracking ability.

The three adaptive neural decoders of Kalman filter (recalib), DyEnsemble and EvoEnsemble obtain higher performance compared with static ones. Kalman filter (recalib) recalibrates the neural decoder every 20 time slots to cope with changes in neural functions [16]; and DyEnsemble increases the model pool with two strategies of connection dropout and weight perturbation [20]. Compared with the Kalman filter (recalib), EvoEnsemble improves the CCs by 8.4%, 14.7%, 2.4% and 10.9% with the four datasets respectively; and the improvement are 4.5%, 6.6%, 15.7% and 3.2% respectively compared with the DyEnsemble approach. The results demonstrate the superior ability of EvoEnsemble in accurate and robust neural decoding with neural functional changes.

**Functional tracking performance.** In Fig. 4 (d), we further illustrate the functional tracking performance of EvoEnsemble with and without the model evolution process. In each subfigure, the red line indicates the function parameter computed in every temporal time slot given neural signals and the true movement trajectories, which we regard as the ground truth of the parameter

Table 1: Decoding performance (CC) of different decoders.

| Decoder | Data-M | Data-P1 | Data-P2 | Data-P3 |
|---|---|---|---|---|
| Particle filter | $0.699 \pm 0.004$ | $0.337 \pm 0.006$ | $0.437 \pm 0.017$ | $0.343 \pm 0.005$ |
| Kalman filter | $0.713 \pm 0.000$ | $0.393 \pm 0.000$ | $0.445 \pm 0.000$ | $0.340 \pm 0.000$ |
| Kalman filter (recalib) [16] | $0.705 \pm 0.000$ | $0.477 \pm 0.000$ | $0.548 \pm 0.000$ | $0.439 \pm 0.000$ |
| DyEnsemble [20] | $0.731 \pm 0.001$ | $0.513 \pm 0.001$ | $0.485 \pm 0.001$ | $0.472 \pm 0.001$ |
| EvoEnsemble (ours) | $\mathbf{0.764 \pm 0.007}$ | $\mathbf{0.547 \pm 0.011}$ | $\mathbf{0.561 \pm 0.017}$ | $\mathbf{0.487 \pm 0.011}$ |

over time. The yellow lines indicate the estimation of the parameters with both EvoEnsemble (right) and EvoEnsemble (w/o DE) (left). Results demonstrate that, although EvoEnsemble (w/o DE) demonstrates its model adaptation ability to a certain extent, the range of functional adjustment is limited with a fixed model pool. Thus EvoEnsemble (w/o DE) can be effective with slight functional changes in a certain range, and face difficulties with substantial changes in functions. On the other hand, the EvoEnsemble model can closely follow the changes in the parameters over time, such that it improves the functional tracking ability as well as the neural decoding accuracy (see Fig. S2 and Fig. S3 for more examples).

## 5 Conclusions

We propose an evolutionary ensemble Bayesian model (EvoEnsemble) for accurate and robust neural decoding under functional changes in neural signals. It incorporates evolutionary computation in a Bayesian filter framework which enables the temporal evolution of neural decoder driven by the model likelihood given incoming neural signals. Experiments with both simulations and neural signals strongly demonstrate the necessity and importance of the functional tracking for robust state estimation with time-varying functions. EvoEnsemble can track the changing functions effectively with both simulation and neural signals to improve the accuracy and robustness in state estimation. One limitation of our approach is the computational efficiency of the evolution process, since it requires hundreds of generations for one model pool update process. Our future work may focus on improving the efficiency of evolutionary computation. The framework of EvoEnsemble is beneficial to a wide range of problems in the nonstationary signal processing area.

## 6 Acknowledgment

This work was partly supported by the grants from National Key R&D Program of China (2018YFA0701400), Key R&D Program of Zhejiang (2022C03011), Natural Science Foundation of China (61906166, U1909202, 61925603), the Fundamental Research Funds for the Central Universities, the Starry Night Science Fund of Zhejiang University Shanghai Institute for Advanced Study (SN-ZJU-SIAS-002), and the Lingang Laboratory (LG-QS-202202-04).

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
