# Supplementary Materials for Tracking Functional Changes in Nonstationary Signals with Evolutionary Ensemble Bayesian Model for Robust Neural Decoding

**Xinyun Zhu**[1,2,4]**, Yu Qi**[3,4,*]**, Gang Pan**[4,2]**, Yueming Wang**[1,4,*]

{zhuxinyun, qiyu, gpan, ymingwang}@zju.edu.cn

[1]Qiushi Academy for Advanced Studies, Zhejiang University, Hangzhou, China
[2]College of Computer Science and Technology, Zhejiang University, Hangzhou, China
[3]Affiliated Mental Health Center & Hangzhou Seventh People's Hospital and the
MOE Frontier Science Center for Brain Science and Brain-machine Integration,
Zhejiang University School of Medicine, Hangzhou, China.
[4]The State Key Lab of Brain-Machine Intelligence, Zhejiang University, Hangzhou, China

## A    Training Subsets Selection

There are many ways to generate the training subsets. In this manuscript, we use the basic one in which the subsets are consecutive, having the same length and being evenly distributed in the training set.

Specifically, suppose the length of the training set is $l_{\text{all}}$. In other words, the training set contains $l_{\text{all}}$ time steps. The training subsets are chosen to be continuous segments, whose start and end indices can be determined through deciding two parameters, namely segment ratio $r_{\text{seg}}$ and segment number $N$. The segment length $l_{\text{seg}}$ can be computed according to the length of the training set $l_{\text{all}}$ and the ratio $r_{\text{seg}}$, where $l_{\text{seg}} = l_{\text{all}} \cdot r_{\text{seg}}$. The segment windows are uniformly distributed over the training time with a stride $l_{\text{stride}} = \lceil \frac{(1-r_{\text{seg}}) \cdot l_{\text{all}}}{N} + \frac{1}{2} \rceil$. Then the segment indices of the $i^{th}$ subset is from $(i-1) \cdot l_{\text{stride}} + 1$ to $\min\{l_{\text{all}}, (i-1) \cdot l_{\text{stride}} + l_{\text{seg}}\}$.

With the indices, we can cut the training data of both neural signals and kinematics and then fit them in the measurement equation to get the initial candidate models.

## B    The Canonical DE

The differential evolution (DE) algorithm is a heuristic global optimization algorithm using a random search method, which does not require gradient information [1, 2]. Unlike traditional genetic algorithms, the individuals in the DE algorithm are encoded in real-value space, and the diversity is achieved through differences between individuals. DE follows an iterative procedure including population initialization, mutation, crossover and selection. There will be a number of iterations from mutation to selection until the best fitness value in the population is good enough or the maximum number of generations set in advance is reached.

### B.1    Initialization

Usually, the parameter to be optimized can be represented by a D-dimension vector $\mathbf{x} = [x_1, x_2, ..., x_D]^T$. And the population containing $N$ individuals, also called target vectors, can be represented by $\{\mathbf{x}_{i,G} = \boldsymbol{x}_{1,i,G}, \boldsymbol{x}_{2,i,G}, ..., \boldsymbol{x}_{D,i,G} | i = 1, 2, ..., N\}$, where $G$ denotes the $G^{th}$ iteration of mutation, crossover and selection. An objective function $f(\mathbf{x}) : \mathbb{R}^D \to \mathbb{R}$, also called

---

*Corresponding authors: Yu Qi and Yueming Wang

36th Conference on Neural Information Processing Systems (NeurIPS 2022).

fitness function in DE is defined for searching the best parameter $\mathbf{x}^*$, where $f(\mathbf{x}^*) > f(\mathbf{x})$ for all $\mathbf{x}$ in the searching space.

In the initial process of DE, a $N$ population, $\{\mathbf{x}_{i,0} = \boldsymbol{x}_{1,i,0}, \boldsymbol{x}_{2,i,0}, ..., \boldsymbol{x}_{D,i,0} | i = 1, 2, ..., N\}$ is sampled randomly from a uniform distribution, where the boundaries are defined as $\mathbf{x}_{\text{low}} = \{\boldsymbol{x}_{1,\text{low}}, \boldsymbol{x}_{2,\text{low}}, ..., \boldsymbol{x}_{D,\text{low}}\}$ and $\mathbf{x}_{\text{up}} = \{\boldsymbol{x}_{1,\text{up}}, \boldsymbol{x}_{2,\text{up}}, ..., \boldsymbol{x}_{D,\text{up}}\}$.

## B.2  Mutation

For each generation $G$, a mutated population $\{\mathbf{v}_{i,G} | i = 1, 2, ..., N\}$ will be created based on the parent population $\{\mathbf{x}_{i,G} | i = 1, 2, ..., N\}$ according to a certain mutation strategy. One of the most popular mutation strategies, "DE/current-to-best/1", is as follows:

$$\mathbf{v}_{i,G} = \mathbf{x}_{i,G} + F \cdot (\mathbf{x}_{\text{best},G} - \mathbf{x}_{i,G}) + F \cdot (\mathbf{x}_{r_i^1,G} - \mathbf{x}_{r_i^2,G}), \tag{1}$$

where the $r_i^1$ and $r_i^2$ are randomly selected indices in the range of $[1, N]$, being distinct from each other and also the index $i$. The $\mathbf{x}_{\text{best},G}$ is the individual who has the best fitness value. The $F$ is a positive factor, used for scaling the difference between the best vector and the current vector, as well as the difference between the randomly selected vectors.

## B.3  Crossover

The final trial vector $\{\mathbf{u}_{i,G} = \boldsymbol{u}_{1,i,G}, \boldsymbol{u}_{2,i,G}, ..., \boldsymbol{u}_{D,i,G} | i = 1, 2, ..., N\}$ is obtained through crossover:

$$\boldsymbol{u}_{j,i,G} = \begin{cases} \boldsymbol{v}_{j,i,G}, & \text{if } \text{rand}(0, 1) \leq CR \text{ or } j = j_{\text{rand}} \\ \boldsymbol{x}_{j,i,G}, & \text{otherwise} \end{cases}, \tag{2}$$

where $j_{\text{rand}}$ is a randomly chosen index from the range $[1, D]$, which ensures that there is at least one dimension of $\mathbf{u}_{i,G}$ comes from the $\mathbf{v}_{i,G}$. And $CR \in [0, 1]$ is a crossover factor, representing the average fraction of vector components that are inherited from the mutation vector.

## B.4  Selection

In the selection stage, the fitness values of the target vectors and the trial vectors are calculated, which is used to determine whether a vector stays in the new population or not. The selected vectors are described as follows:

$$\mathbf{x}_{i,G+1} = \begin{cases} \mathbf{u}_{i,G}, & \text{if } f(\mathbf{u}_{i,G}) > f(\mathbf{x}_{i,G}) \\ \mathbf{x}_{i,G}, & \text{otherwise} \end{cases}, \tag{3}$$

where the $f(\cdot)$ represents the fitness function in a maximum problem.

The iteration of mutation, crossover and selection will be terminated when it reaches the pre-defined number of generation, or satisfies the pre-specified fitness value. Also, an early-stopping strategy can be adopted. That is, if the best fitness among the population no longer increases significantly during several successive iterations, the process will be stopped advancely.

# C  Apply JaDE to the Evolution Process

In the context of the model evolution in neural decoding, the data that DE deals with can be particularly different in each update process. It is difficult to cover all situations with a fixed set of parameters, including the scaling factor $F$ and the crossover probability $CR$. Improper parameters will lead to stagnation or premature convergence [3]. Therefore, we apply JADE [4] to our evolution process, which adjusts $F$ and $CR$ adaptively. Besides, following the JADE, we adopt the mutation strategy "DE/current-to-pbest/1" with archive.

## C.1  "DE/current-to-pbest/1" Strategy

To avoid the premature convergence problem and diversify the population, we adopt a mutation strategy of "DE/current-to-pbest/1":

$$\mathbf{v}_{i,G} = \mathbf{x}_{i,G} + F_i \cdot (\mathbf{x}_{\text{pbest},G} - \mathbf{x}_{i,G}) + F_i \cdot (\mathbf{x}_{r_i^1,G} - \tilde{\mathbf{x}}_{r_i^2,G}), \tag{4}$$

where $\mathbf{x}_{\mathrm{pbest},G}$ represents the individual randomly selected from the top $100 \times p\%$ population. $r_i^1$ and $r_i^2$ are randomly selected indices in the range of $[1, N]$, being distinct from each other and also the index $i$.

## C.2 Adaptive Mutation Factor Selection

The $F$ in Eq. (4) is a list of the positive mutation factor. In each generation $G$, the list is generated from a Cauchy distribution with the initial mutation factor $\mu_F$ and a scale parameter 0.1:

$$F = \mathrm{randc}(\mu_F, 0.1), \tag{5}$$

setting the maximum value to be 1 and the minimum value to be 0. When a generation finishes, the list will be updated according to all the successful mutation factors in this generation, which is denoted as $S_F$:

$$\mu_F = (1 - c) \cdot \mu_F + c \cdot \mathrm{mean}_L(S_F), \tag{6}$$

where $c \in (0, 1]$ is a ratio to control the effect of current generation and $\mathrm{mean}_L(S_F)$ represents the Lehmer mean:

$$\mathrm{mean}_L(S_F) = \frac{\sum_{F \in S_F} F^2}{\sum_{F \in S_F} F}. \tag{7}$$

## C.3 "DE/current-to-pbest/1" with Archive

The $\tilde{\mathbf{x}}_{r_i^2,G}$ in Eq. (4) is randomly selected from a collection storing the inferior individuals in the previous selection operation. Specifically, in the selection process of each generation, the individual after mutation and crossover whose fitness value is smaller than the original individual will fail to be selected into the population in the next generation. We add these inferior individuals to a external archive $\mathbf{A}$ as candidate individuals in the latter mutation process. Suppose current population is denoted by $\mathbf{P}$, then the $\tilde{\mathbf{x}}_{r_i^2,G}$ is randomly selected from the union $\mathbf{A} \cup \mathbf{P}$.

## C.4 Adaptive Crossover Factor Selection

The individual $\{\mathbf{u}_{i,G} = \boldsymbol{u}_{1,i,G}, \boldsymbol{u}_{2,i,G}, ..., \boldsymbol{u}_{D,i,G} | i = 1, 2, ..., N\}$ after crossover is obtained through:

$$\boldsymbol{u}_{j,i,G} = \begin{cases} \boldsymbol{v}_{j,i,G}, & \text{if } \mathrm{rand}(0,1) \leq CR_i \text{ or } j = j_{\mathrm{rand}} \\ \boldsymbol{x}_{j,i,G}, & \text{otherwise} \end{cases}, \tag{8}$$

where $j_{\mathrm{rand}}$ is a randomly chosen index from the range $[1, D]$, which ensures that there is at least one dimension of $\mathbf{u}_{i,G}$ comes from the $\mathbf{v}_{i,G}$. The $CR$ is a list of the positive crossover factor, which is similar to $F$. Specifically, in each generation $G$, the list is generated from a normal distribution:

$$CR = rand(\mu_{CR}, 0.1), \tag{9}$$

setting the maximum value to 1 and the minimum value to 0. The update formulation of CR is:

$$\mu_{CR} = (1 - c) \cdot \mu_{CR} + c \cdot \mathrm{mean}_A(S_{CR}), \tag{10}$$

where $c \in (0, 1]$ is a the same ratio in Eq. (6), and $S_{CR}$ is all the successful crossover factors. The $\mathrm{mean}_A(S_{CR})$ represents the arithmetic mean as usual.

# D More Details

## D.1 More Details in the Procedure of EvoEnsemble

The EvoEnsemble approach consists of two phases, one is the calibration phase and the other is the test phase.

During the calibration phase, training sample pairs $(\boldsymbol{x}_t, \boldsymbol{y}_t)$ are collected and used to learn the transition function $g(\cdot)$, as well as a set of initial models $m^k(\cdot)$ to approximate the time-varying observation function $h_t(\cdot)$. Once the calibration phase is over, the transition function $g(\cdot)$ is fixed. Meanwhile, a set of initial particles $\{\boldsymbol{x}_t^{(1)}, \boldsymbol{x}_t^{(2)}, ..., \boldsymbol{x}_t^{(N_{\mathrm{par}})}\}$ are randomly drown from a Gaussian distribution, to approximate the posterior of the kinematic states $p(\boldsymbol{x}_t | \boldsymbol{y}_{0:t})$.

During the test phase, the observation function $h_t(\cdot)$ changes along with the incoming neural signals $\boldsymbol{y}_t$ through adaptively assembling the models $m^k(\cdot)$ in the model pool $\mathcal{M}$. Note that, the model weights $w_t^k$ are updated at each time slot, and the model parameters $\mathbf{p}^k$ are updated every several time slots, which is the most critical point of EvoEnsemble.

Specifically, we use particle filter algorithm to implement this process. And there are several steps to recursively estimate the posterior of kinematic states $p(\boldsymbol{x}_t|\boldsymbol{y}_{0:t})$:

Step 1: For each particle $\boldsymbol{x}_t^{(s)}$, use the kinematic posterior $p(\boldsymbol{x}_{t-1}|\boldsymbol{y}_{0:t-1})$ at time $t-1$ and the transition function $g(\cdot)$ to estimate the kinematic prior $p(\boldsymbol{x}_t^{-}|\boldsymbol{y}_{0:t})$ at time $t$ .

Step 2: Use the kinematic prior $p(\boldsymbol{x}_t^{-}|\boldsymbol{y}_{0:t})$ at time $t$ and the encoding models $m^k(\cdot)$ to estimate corresponding neural signals $\hat{\boldsymbol{y}}_t^k$

Step 3: Compute likelihood $p^{k(s)}(\boldsymbol{y}_t|\boldsymbol{y}_{0:t-1})$ for each particle in each model $m^k(\cdot)$ by comparing the encoded neural signals $\hat{\boldsymbol{y}}_t^k$ with the observed neural signals $\boldsymbol{y}_t^k$. The model likelihood $p^k(\boldsymbol{y}_t|\boldsymbol{y}_{0:t-1})$ is equal to the sum of its particle likelihood.

Step 4: Estimate the state posterior $p^k(\boldsymbol{x}_t|\boldsymbol{y}_{0:t})$ calculated by each model through taking a weighted sum of its particles. Compute the final state posterior $p(\boldsymbol{x}_t|\boldsymbol{y}_{0:t})$ through taking a weighted sum of all models' estimation.

When all models' likelihoods in the model pool are small, the model pool will be updated by EvoEnsemble through a differential evolution algorithm, where model fitness is equal to the model likelihood.

### D.2 More Details in Particle-based Solution

We randomly sample particles $x_t^{(s)}$ from a standard Gaussian distribution. At each time slot, they are estimated one step ahead through transition function $g(\cdot)$. For each encoding model $m^k(\cdot)$, its corresponding particle likelihoods are calculated by comparing the encoded neural signals $\hat{\boldsymbol{y}}_t^k$ with the observed neural signals $\boldsymbol{y}_t^k$. Here, we assume the single observation functions are linear, and the distribution of the state and observation variables are multivariate Gaussian. Then for each model $m^k(\cdot)$, the single observation equation can be described as follows:

$$\boldsymbol{y}_t = \mathbf{M}^k\boldsymbol{x}_t + v_t, \tag{11}$$

where $v_t \sim \mathcal{N}(0, Q)$. Then the importance weights for each particle of each model can be calculated by:

$$\omega_t^{k(s)} = p(\hat{\boldsymbol{y}}_t^k|\boldsymbol{x}_t^{(s)}) = (2\pi)^{-d_y/2}|Q|^{-1/2}\exp\{-1/2(\hat{\boldsymbol{y}}_t^k - \mathbf{M}\boldsymbol{x}_t^{(s)})'Q^{-1}(\hat{\boldsymbol{y}}_t^k - \mathbf{M}\boldsymbol{x}_t^{(s)})\}. \tag{12}$$

### D.3 More Details in Simulation Data

The kinematics $\boldsymbol{x}_t$ is randomly generated in the range of $[0, 1]$ and is smoothed with a window size of 20. The measurement in the training data is generated by the fixed mapping matrix $[h_0^1, h_0^2]^{\mathrm{T}}$, while the measurement in the test data $y_t$ is generated by a continuous changing mapping matrix $[h_t^1, h_t^2]^{\mathrm{T}}$. Random Gaussian noise $N(0, 0.01)$ is added to the measurements. "E" is the representation in scientific notation.

### D.4 More Details in Clinical Data

"Ortho-impedance" is a commonly used computer assistant way in the brain-machine interface, which reserves the projection of the control velocity in the ideal direction, i.e., the direction from the cursor to the target, and decreases the projection of control velocity in perpendicular to the ideal direction, preventing deviation from the target direction.

# E  Parameter Configurations

## E.1  Simulations

We fix the parameters for all five conditions. We use a model pool with $N = 50$ models, and each model is fit on a subset of training set, where the segment ratio $r_{\text{seg}}$ is 0.1. The parameters in JaDE are set to $p = 0.1$, $c = 0.05$, $\mu_F = 0.1$ and $\mu_{CR} = 0.1$. We adopt a regular update strategy with update interval $t_{\text{up}} = 15$. The max generation of DE is set to 100, with an early-stopping operation if the fitness do not decrease significantly for 10 iterations. The window length of the previous time steps we used for one time update $l_{\text{pre}}$ is 30. The preserve ratio in history-model-archive strategy is 0.8.

## E.2  Data-M

For Data-M, We use a model pool with $N = 20$ models, and each model is fit on a subset of training set, where the segment ratio $r_{\text{seg}}$ is 0.5. The parameters in JaDE are set to $p = 0.2$, $c = 0.05$, $\mu_F = 0.2$ and $\mu_{CR} = 0.1$. We both adopt regular and adaptive update strategy to achieve better performance, with update interval $t_{\text{up}} = 15$ and update ratio $r_{\text{up}} = 2/3$. The max generation of DE is set to 300, with an early-stopping operation if the fitness do not decrease significantly for 20 iterations. The window length of the previous time steps we used for one time update $l_{\text{pre}}$ is 15. The preserve ratio in history-model-archive strategy is 0.5.

## E.3  Data-P

For Data-P1/P2/P3, We use a model pool with $N = 20$ models, and each model is fit on a subset of training set, where the segment ratio $r_{\text{seg}}$ is 0.5. The parameters in JaDE are set to $p = 0.2$, $c = 0.05$, $\mu_F = 0.3$ and $\mu_{CR} = 0.1$. We both adopt regular and adaptive update strategy to achieve better performance, with update interval $t_{\text{up}} = 15$ and update ratio $r_{\text{up}} = 0.4$. The max generation of DE is set to 300, with an early-stopping operation if the fitness do not decrease significantly for 20 iterations. The window length of the previous time steps we used for one time update $l_{\text{pre}}$ is 15. The preserve ratio in history-model-archive strategy is 0.3.

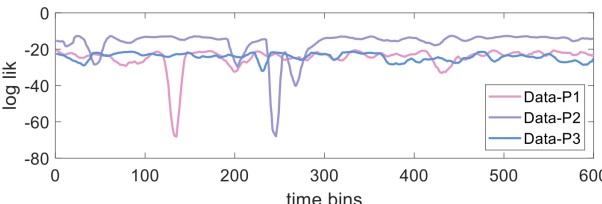

Figure S1: The changes of log-likelihood on the three clinical datasets. EvoEnsemble improves most on the dataset which the likelihood drops most.

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

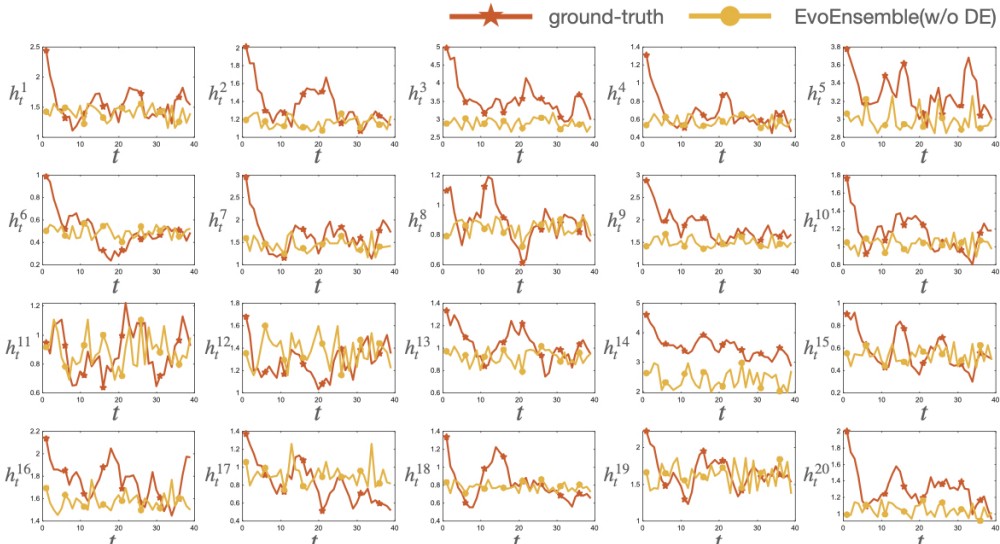

Figure S2: Tracking performance of EvoEnsemble(w/o DE) on 20 mapping parameters in Data-M. The horizontal axis t denotes the $t^{th}$ update in EvoEnsemble. The red line is the ground-truth which we fit on the test dataset while the yellow one is the assembled parameters from the model pool of EvoEnsemble(w/o DE).

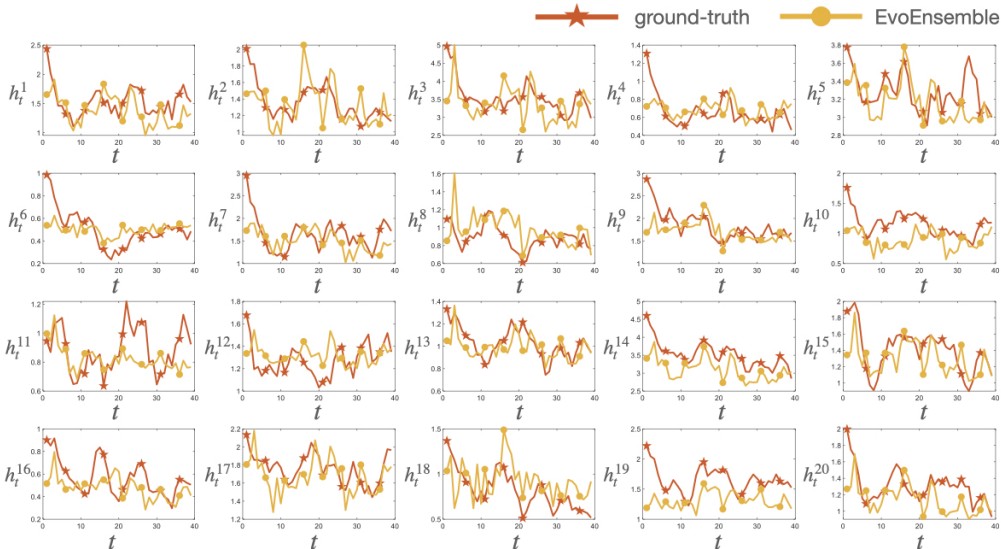

Figure S3: Tracking performance of EvoEnsemble on 20 mapping parameters in Data-M.