# OpenReview forum: "Tracking Functional Changes in Nonstationary Signals with Evolutionary Ensemble Bayesian Model for Robust Neural Decoding"
_NeurIPS.cc/2022/Conference — NeurIPS 2022 Accept_

### Official Review · Reviewer_3dMs · 2022-06-17

**Rating:** 5
**Confidence:** 2
**Soundness:** 3 good
**Presentation:** 2 fair
**Contribution:** 3 good

**Summary:**

The paper proposes a method for adaptively updating neural decoders in Brain Machine Interfaces (BMI). Evidently the method consists in maintaining a population of decoders, and evolving them according to a genetic algorithm, with fitness determined by the likelihood of the decoders given the data. Various experiments purport to demonstrate the superior performance of the method and the effects of ablating various components.


**Questions:**

1- Line 71, we are told that we want to use the neural data y(t) to predict certain kinematic values x(t), e.g. a cursor position. Yet the whole method apparently consists in training decoders for the opposite task, i.e. predicting neural values y(t) from the kinematics x(t). Please explain this choice, and why we couldn't simply train y -> x predictors. It may be obvious to you, but not to a general reader.

2- In Step 1 (line 111) we are told that we get the "prior kinematics" x(t) from x(t-1) through the "transition function" g. What is this transition function? If it already gives us x(t), what is the need of predicting it in the first place?

3-  In Step 2, we supposedly use the decoders to predict neural signals y_hat / x(t), where x(t) comes from the "prior kinematics" from Step 1. But when I look at Equation 4, and in fact at all other equations in the paper, the estimated distribution over x(t) is instead computed from the previous y (presumably with the "particle system" of section 3.3)? So which one is actually used to estimate x(t)? Where are the "prior kinematics" used exactly?

4- In Eq 4, it seems that the likelihood of each decoder (p^k) is computed by marginalizing over the possible kinematics x(t), weighted by their *estimated* probability given previous y. We are told "a particle solution towards it" in Section 3.3. Towards what?

5- In section 3.3, apparently we should learn how p(x(t) / y(t'<t)) is estimated for any x. Unfortunately this section is incomprehensible. We are told of "particles" x1...xn (chosen how?), with "important weights" (would that be "importance weights"?)  called omega^k(s)_t... Where do these omegas come from? What do they represent? How are they computed?


6- More generally: x(t) is estimated from the decoders (Equation 11). Decoders are evaluated based on their output for estimated x(t). This seems circular. At which point does *real* data about x(t) enter the system? AFAICT, in the paper, every single value of x(t) used in the equations is estimated,  with nothing to anchor it to real data.  So the predictions could be absolutely random, as long as the decoders can use them to predict neural data y(t) accurately? Surely this is not the case! I suspect some part of the description is missing.

7- How are the model predictions  y-hat(t) compared with real neural data y(t)? That is, how is p^k(y_t / t_t) (in Equation 4, right-hand side, first factor in the integral) actually computed?

8- The experiments are similarly lacking in description:

- In section 4.1 (simulations) what is h_t? How are the x(t) and the y(t) generated? Also, what is "E" ?
- In section 4.2 (monkey reaching) what is being tracked?
- The description of the human clinical dataset is particularly lacking. What is the x(t) being tracked? What is an "ortho-impedance"?

I note that in all of these, the "transition function" g(t) is never mentioned, leaving me confused about  what exactly it represent.


9- More pressingly for the actual impact of the method, it is briefly mentioned that the evolutionary algorithm is run for "hundreds of generations" (line 149). How long does that take? Does it mean that the method cannot be applied online to real-time data? That would be a serious limitation.



**Limitations:**

The authors briefly mention that their method requires running an algorithm for hundreds of generations, for every update of the decoder pool (?). However, they do not quantify the time taken. Most importantly, they do not say whether this prevents the method from being applied online, in real-time, in "live" BMIs (as opposed to the offline datasets used in the experiments).

Please explicitly state whether the method is currently applicable to online, real-time neural data from live BMIs or not.

Social impact does not seem relevant in the current state of the method.

**Strengths And Weaknesses:**


The method seems potentially interesting. Unfortunately the extremely confused description makes it impossible to understand, much less reproduce the method. This applies both to the description of the method itself, as well as the experiments, as explained below.

UPDATE: After the author's response and clarifications, the paper is much improved and I have updated my score towards acceptance.

---

> ### Author Response · Authors · 2022-08-02
> **Response to Reviewer 3dMs**
>
> Thank you for the valuable comments and feedback. We have given a detailed point-to-point response to your comments below. Hopefully these will resolve most of your concerns, and can be taken into consideration when deciding on the final review score of the paper.
>
> 1. ***"Why we couldn't simply train $y \rightarrow x$ predictors?"***
> - Predicting neural values $y_t$ from the kinematics $x_t$, namely the neural encoders, are important in understanding how our brain represents information such as movements. In neural decoding, there are two typical kinds of decoders: direct regression, which is a $y \rightarrow x$ predictor; and encoder-based decoder, which aims to decode information with the guidance of encoding models [[Inoue et al., 2018](https://www.nature.com/articles/s41467-018-07647-3.pdf?origin=ppub)]. Although direct regression can be more simple and straightforward, existing approaches tend to use the encoder-based decoders for better explainable features, including population vector algorithm, Kalman filters and Bayesian filters [[Gilja et al., 2015](https://www.ncbi.nlm.nih.gov/pmc/articles/PMC4805425/pdf/nihms761686.pdf)] [[Nuyujukian et al., 2018](https://journals.plos.org/plosone/article/file?id=10.1371/journal.pone.0204566&type=printable)]. The proposed EvoEnsemble is an encoder-based decoder approach which therefore includes an encoding model inferring $y_t$ given $x_t$.
>
> 2. ***"In Step 1, what is the transition function $g(\cdot)$? If it already gives us $x_t$, what is the need of predicting it in the first place?"***
> - The meaning of transition function $g(\cdot)$ in our work is consistent with the classic state-space model [[Welch et al., 1995](https://perso.crans.org/club-krobot/doc/kalman.pdf)]. There is one thing needs to be clear that we have training and test phases. Here, the transition function $g(\cdot)$ is actually a linear mapping matrix, describing the transfer process of $x_{t-1}$ to $x_t$, which can be learned from the training set. The test phase is a recursive process, and the initial state $x_0$ can be randomly generated. In Step 1, all we have to do is use the learned transition function $g(\cdot)$ and the previous state $x_{t-1}$ to estimate the current state $x_t$ as shown in Eq. (2).
>
> 3. ***"Which one is actually used to estimate $x_t$? Where are the "prior kinematics" used exactly?"***
> - As we explained above, the state $x_t$ is recursively estimated from the prior state $x_{t-1}$ using the state transition function $g(\cdot)$. And the initial state $x_0$ is randomly sampled from a Gaussian distribution.
>
> 4. ***"A particle solution towards what?"***
> - We mean a particle solution towards Eq. (4). Note that Eq. (4) is the formal expression of model likelihood while Eq. (12) is the solution to it using the classic particle filter algorithm [[Arulampalam et al., 2002](http://www.dcsc.tudelft.nl/~sc4081/2018/assign/pap/andrea_PFR_paper.pdf)].
>
> 5. ***"How to choose "particles"? What do omegas come from? What do they represent? How are they computed?"***
> - The particles are randomly sampled from a Gaussian distribution at the initial moment, and then transferred through the state transition function $g(\cdot)$. The omegas are importance weights, which are computed through the likelihood of $y_t$ given $x_t^{(s)}$ as traditional particle filter does [[Doucet et al., 2000](https://www.irisa.fr/aspi/legland/ensta/ref/doucet00b.pdf)][[Elfring et al., 2021](https://www.mdpi.com/1424-8220/21/2/438/pdf)].
>
> 6. ***"At which point does real data about $x_t$ enter the system? AFAICT, in the paper, every single value of $x_t$ used in the equations is estimated, with nothing to anchor it to real data. So the predictions could be absolutely random, as long as the decoders can use them to predict neural data y(t) accurately?"***
> - The prediction is not absolutely random. Again, this is a recursive process and all we need to do is to set the initial particle state $x_0^{(1)},x_0^{(2)},...,x_0^{(N_{\text{par}})}$, and then the observation $y_t$ will help to correct the state.

---

> > ### Author Response · Authors · 2022-08-02
> > **Additional Response to Reviewer 3dMs**
> >
> > 7. ***"How are the model predictions $\hat{y}_t$ compared with real neural data $y_t$?"***
> > - This is the same question as "where do omegas come from". More specifically, we first learn the distribution of the estimation residual from the training data, and then query the probability of estimation residual in the test phase under the distribution learned from the training phase.
> >
> > 8. ***"In section 4.1 (simulations) what is $h_t$? How are the $x_t$ and the $y_t$ generated? Also, what is “E” ?"***
> > - $h_t$ is the changing function we want to track. That is, the changing mapping matrix. The kinematics $\boldsymbol{x}_t$ is randomly generated in the range of $[0,1]$ and is smoothed with a window size of 20. The measurement in the training data is generated by the fixed mapping matrix $[h_0^1,h_0^2]^\text{T}$, while the measurement in the test data $y_t$ is generated by a continuous changing mapping matrix $[h_t^1,h_t^2]^\text{T}$. Random Gaussian noise $N(0,0.01)$ is added to the measurements. "E" is the representation in scientific notation.
> >
> > 9. ***"In section 4.2 (monkey reaching) what is being tracked?"***
> > - The changing function, i.e., the parameter of the encoding matrix is being tracked.
> >
> > 10. ***"What is the $x_t$ being tracked? What is an “ortho-impedance”?"***
> > - We are not tracking $x_t$, but $h_t$. "ortho-impedance" is a commonly used computer assistant way in the brain-machine interface, which reserves the projection of the control velocity in the ideal direction, i.e., the direction from the cursor to the target, and decreases the projection of control velocity in perpendicular to the ideal direction, preventing deviation from the target direction [[Collinger et al., 2013](https://www.ncbi.nlm.nih.gov/pmc/articles/PMC3641862/pdf/nihms448153.pdf)] [[Qi et al., 2022](https://ieeexplore.ieee.org/stamp/stamp.jsp?tp=&arnumber=9795216)].
> >
> > 11. ***"How long does evolutionary algorithm takes? Can it be applied online to real-time data?"***
> > - The computational costs of one model evolution generation process are $1.130 \pm 	0.172$ s, $0.489 \pm 	0.109$ s, $0.519 \pm 0.122$ s, $0.509 \pm 	0.120$ s, for Data-M, Data-P1, Data-P2 and Data-P3 respectively. With the evolve-at-changes strategy, model evolution is only perform-as-needed, which largely decrease the computational costs. Parallel computing-based implementation can further speed up the computation. Specifically, we can separate the model evolution and the dynamic model assembling in two processes, where the model pool update module runs silently in the background, collecting data and evolving models. Once it finishes updating, replace the original model pool. Such parallel operations can enable online real-time decoding.

---

> > ### Comment · Reviewer_3dMs · 2022-08-07
> > **Response**
> >
> > Thank you for your response.
> >
> > Based on the response and the new version of the paper, my understanding of the proposed method is as follows:
> >
> > - Using separate training data, we first train $g(x(t-1)) = x(t)$, as well as a bunch of models (approximations of $h$) $m_k$.
> >
> > - At any time step, we predict a prior on $x(t)$, using $g$ (but what is the $x(t-1)$ being used??)
> >
> > - We update the likelihood of each $m_k$, based on the match between their predicted $y(t) = m_k(x(t))$ (using the predicted $x(t)$) and the real observed $y(t)$.
> >
> > - We use this likelihood to update the posterior probability of each model $m_k$ being the right $h_t$, and then use *that* to estimate a posterior probability on $x(t)$ through a particle method.
> >
> > - When performance degrades, we apply a differential evolution algorithm to update the $m_k$, using their likelihoods as fitness (why not use the posterior?)
> >
> > If this summary is reasonably accurate, please do include a similar summary at the beginning of section 2, so that readers not intimately familiar with this field have a chance of following the complex description that follows.
> >
> > The main remaining problems with the new version are:
> >
> > 1- The particle system in Section 3.3 is woefully under-described. Please do explain how the Omegas are computed. That is, include an equation with Omega^k(s)_t = on the left side. Explain also where the particles x^s_t come from - how are they chosen, when and how are they updated?
> >
> > 2- From equation 13, it appears that p(x_t) has only discrete support, i.e. it is only nonzero for the particles maintained by the system. How can we use this to compute an actual prediction for x(t) ? How can we compare this discrete p(x(t)) with actual data, where the observed x(t) will usually have p=0 under this discrete distribution?
> >
> > 3- L. 103: We "infer" the probability distribution of the x(t) by using g. But g is a deterministic linear mapping, taking a single vector as input. How can it be used to produce a distribution? What is x(t-1), exactly? Is it the whole set of "particles" described in Section 3.3? If so, this should be stated explicitly.
> >
> > 4- L. 164: What is the "Bayesian filter"? Is it the particle  system described in the next paragraph? If so it should be stated!
> >
> > 5- Why use likelihoods rather than posteriors (eq. 10) as the evolutionary fitness of the models? (The answer may be obvious, but it escapes me right now.)
> >
> > Minor things:
> >
> > l. 166: Again, I believe "important weights" should be "importance weights".
> > l.20: ".Because" -> ", because".

---

> > > ### Author Response · Authors · 2022-08-09
> > > **Additional Response to Reviewer 3dMs - ( part one )**
> > >
> > > Many thanks for your constructive suggestions and feedback. We have revised the manuscript accordingly and given a detailed point-to-point response to your comments below. Hopefully, these will address most of your concerns. If you have further questions, please do not hesitate to reply to us.
> > >
> > > ***"Based on the response and the new version of the paper, my understanding of the proposed method is as follows:***
> > > ***· Using separate training data, we first train $g(x(t-1))=x(t)$, as well as a bunch of models (approximations of $h$) $m_k$.***
> > > ***· At any time step, we predict a prior on $x(t)$, using $g$ (but what is the $x(t-1)$ being used??)***
> > > ***· We update the likelihood of each $m_k$, based on the match between their predicted $y(t)=m_k(x(t))$ (using the predicted $x(t)$) and the real observed $y(t)$.***
> > > ***· We use this likelihood to update the posterior probability of each model $m_k$ being the right $h_t$, and then use that to estimate a posterior probability on $x(t)$ through a particle method.***
> > > ***· When performance degrades, we apply a differential evolution algorithm to update the $m_k$, using their likelihoods as fitness (why not use the posterior?).***
> > >
> > > ***If this summary is reasonably accurate, please do include a similar summary at the beginning of section 2, so that readers not intimately familiar with this field have a chance of following the complex description that follows."***
> > >
> > > - The summary is generally reasonably accurate. And we have refined this summary as follows:
> > > > - The EvoEnsemble approach consists of two phases, one is the calibration phase and the other is the test phase.
> > > > - During the calibration phase, training sample pairs $(x_t,y_t)$ are collected and used to learn the transition function $g(\cdot)$, as well as a set of initial models $m^k(\cdot)$ to approximate the time-varying observation function $h_t(\cdot)$.
> > > > - Once the calibration phase is over, the transition function $g(\cdot)$ is fixed. Meanwhile, a set of initial particles $\{x_t^{(1)},x_t^{(2)},...,x_t^{(N_{\text{par}})}\}$ are randomly drown from a Gaussian distribution, to approximate the posterior of the kinematic states $p(x_t|y_{0:t})$.
> > > >  - During the test phase, the observation function $h_t(\cdot)$ changes along with the incoming neural signals $y_t$ through adaptively assembling the models $m^k(\cdot)$ in the model pool $\mathcal{M}$. Note that, the model weights $w^k_t$ are updated at each time slot, and the model parameters $\mathbf{p}^k$ are updated every several time slots, which is the most critical point of EvoEnsemble.
> > > > - Specifically, we use particle filter algorithm to implement this process. And there are several steps to recursively estimate the posterior of kinematic states $p(x_t|y_{0:t})$:
> > > > - Step 1: For each particle $x_t^{(s)}$, use the kinematic posterior $p({x_{t-1}}|y_{0:t-1})$ at time $t-1$ and the transition function $g(\cdot)$ to estimate the kinematic prior $p({x_t}^-|y_{0:t})$ at time $t$ .
> > > > - Step 2: Use the kinematic prior $p({x_t}^-|y_{0:t})$ at time $t$ and the encoding models $m^k(\cdot)$ to estimate corresponding neural signals $\hat{y}^k_t$。
> > > > - Step 3: Compute likelihood $p^{k(s)}(y_t|y_{0:t-1})$ for each particle in each model $m^k(\cdot)$ by comparing the encoded neural signals $\hat{y}^k_t$ with the observed neural signals $y^k_t$. The model likelihood $p^k(y_t|y_{0:t-1})$ is equal to the sum of its particle likelihood.
> > > > - Step 4: Estimate the state posterior $p^k(x_t|y_{0:t})$ calculated by each model through taking a weighted sum of its particles. Compute the final state posterior $p(x_t|y_{0:t})$ through taking a weighted sum of all models' estimation.
> > > > - When all models' likelihoods in the model pool are small, the model pool will be updated by EvoEnsemble through a differential evolution algorithm, where model fitness is equal to the model likelihood.

---

> > > > ### Comment · Reviewer_3dMs · 2022-08-09
> > > > **Thank you**
> > > >
> > > > I appreciate the author's responses. If these clarifications are included into the paper, I will definitely increase my score.

---

> > > > > ### Author Response · Authors · 2022-08-10
> > > > > **We have put these clarifications into the paper, thank you so much!**
> > > > >
> > > > > Thank you for your advice! We have put these clarifications into the original manuscript as much as possible. Due to space limitations, we put the remaining parts in the appendix. For details, please refer to the Appendix D.

---

> > > ### Author Response · Authors · 2022-08-09
> > > **Additional Response to Reviewer 3dMs - ( part two )**
> > >
> > > Below, we have given a detailed point-to-point response to your remaining problems.
> > >
> > > 1. ***"The particle system in Section 3.3 is woefully under-described. Please do explain how the Omegas are computed. That is, include an equation with $\omega^{k(s)}_t =$ on the left side. Explain also where the particles $x^s_t$ come from - how are they chosen, when and how are they updated?"***
> > >
> > > - We randomly sample particles $x_t^{(s)}$ from a standard Gaussian distribution. At each time slot, they are estimated one step ahead through transition function $g(\cdot)$.
> > > - For each encoding model $m^k(\cdot)$, its corresponding particle likelihoods are calculated by comparing the encoded neural signals $\hat{\boldsymbol{y}}^k_t$ with the observed neural signals ${\boldsymbol{y}}^k_t$.
> > > - Here, we assume the single observation functions are linear, and the distribution of the state and observation variables are multivariate Gaussian. Then the single observation equation can be described as follows: $\boldsymbol{y}_t=\mathbf{M}\boldsymbol{x}_t+v_t$, where $v_t \sim  \mathcal{N}(0,Q)$.
> > > - Then the importance weights for each particle of each model can be calculated by:
> > > $\omega^{k(s)}_t = p(\hat{\boldsymbol{y}}^k_t|\boldsymbol{x}^{(s)}_t)=(2\pi)^{-d_y/2}|Q|^{-1/2}\exp{\{-1/2(\hat{\boldsymbol{y}}^k_t-\mathbf{M}\boldsymbol{x}^{(s)}_t)'Q^{-1}(\hat{\boldsymbol{y}}^k_t-\mathbf{M}\boldsymbol{x}^{(s)}_t)\}}$.
> > >
> > > 2. ***"From equation 13, it appears that $p(x_t)$ has only discrete support, i.e. it is only nonzero for the particles maintained by the system. How can we use this to compute an actual prediction for $x(t)$ ? How can we compare this discrete $p(x(t))$ with actual data, where the observed $x(t)$ will usually have $p=0$ under this discrete distribution?"***
> > >
> > > - Equation (13) is indeed an approximation where $p(x_t|h_t(\cdot)=m^k(\cdot),y_{0:t}) \approx \sum_{s=1}^{N_{\text{par}}}\omega_{t}^{k(s)}\delta(x_t-x_t^{(s)})$. When the $N_{\text{par}} \rightarrow \infty$, the approximation approaches the true posterior density [[Arulampalam et al., 2002](http://www.dcsc.tudelft.nl/~sc4081/2018/assign/pap/andrea_PFR_paper.pdf)].
> > >
> > > 3. ***"L. 103: We "infer" the probability distribution of the $x(t)$ by using $g$. But $g$ is a deterministic linear mapping, taking a single vector as input. How can it be used to produce a distribution? What is $x(t-1)$, exactly? Is it the whole set of "particles" described in Section 3.3? If so, this should be stated explicitly."***
> > >
> > > - For each particle $x_{t-1}^{(s)}$, we can use $g(\cdot)$ to estimate a transferred particle $x_{t}^{(s)}$. The probability distribution $x_t$ is approximated by a number of particles.
> > >
> > > 4. ***"L. 164: What is the "Bayesian filter"? Is it the particle system described in the next paragraph? If so it should be stated!"***
> > >
> > > - Recursive Bayesian filter, also known as Bayes filter, is a general probabilistic approach for estimating an unknown probability density function recursively over time using incoming measurements and a mathematical process model.
> > > - Here, the "Bayesian filter" refers to the whole framework for estimating the kinematic states given the extended state-space model in equation (2) and (3).
> > >
> > > 5. ***"Why use likelihoods rather than posteriors (eq. 10) as the evolutionary fitness of the models? (The answer may be obvious, but it escapes me right now.)"***
> > >
> > > - We can use posteriors as the evolutionary fitness. However, the posterior is related to the prior, which is less direct than the likelihood to reflect the quality of the model.

---

### Official Review · Reviewer_Lhh8 · 2022-07-11

**Rating:** 4
**Confidence:** 4
**Soundness:** 3 good
**Presentation:** 3 good
**Contribution:** 2 fair

**Summary:**

This paper proposed a neural decoding method to process non-stationary data for brain-computer interface, called  “EvoEnsemble”.
The main advantage of the method is that the model can be adaptive to the variation of neural signals through evolution in the Bayesian framework.

**Questions:**

1) The DE algorithm tends to evolve all models towards the same one,  is this evolution approach really better than a well-defined model pool, considering we may can utilize the statistical features of neural signals to well define a good model pool?
2) How about we restrict models in the pool to be disentangled  (such as orthogonal base functions), will this works better?
3) How to understand Fig.2 condition 3, the h^2_t tracking result is quite different from the ground truth, although the likelihood and R^2 are high?
4) In neural decoding tasks, the trend of state x is essential for the application. Still, the decoding results, which always seem to fluctuate up and down with respect to the ground truth, influence the decoding effect.
5) Fig. 2(a) the vertical coordinates should be “h^2_t” rather than “h^1_t”
6) In Eq(9) “\sum_{i=1}^{N}…” should be “\sum_{k=1}^{N}…”
7) In Eq(11) “\sum_{i=1}^{N}…” should be “\sum_{k=1}^{N}…”


**Limitations:**

The authors mentioned the limitation of computational efficiency, but did not evaluate it.

**Strengths And Weaknesses:**

Strengths:
1) The simulation experiments support that the method can capture the non-stationary characteristic of neural signals.
2) The paper is clearly written and well-organized, and the technical contents are well explained.

Weaknesses
1)  The method is built upon the previous work, DyEnsemble (Qi et al. 2019) by introducing evolution of the model pool,  which is OK, but the novelty is not strong under the normal criterion of NeurIPS.
2) The differential evolution (DE) algorithm tends to evolve models towards the best one, even though each candidate model evolves independently. This may make the model pool eventually becomes useless..
3) The computational efficiency of evolutionary computation is not evaluated in the paper, which is key for an online decoding method.

---

> ### Author Response · Authors · 2022-08-02
> **Response to Reviewer Lhh8**
>
> Many thanks for the helpful comments and feedback. We have given a detailed point-to-point response to your comments below. Hopefully these will address most of your concerns, and that they can be taken into consideration when deciding the final score of the paper.
>
> 1. ***“The method is built upon the previous work, DyEnsemble [[Qi et al., 2019](https://proceedings.neurips.cc/paper/2019/file/3f7bcd0b3ea822683bba8fc530f151bd-Paper.pdf)] by introducing evolution of the model pool, which is OK, but the novelty is not strong under the normal criterion of NeurIPS.”***
> - Our work mainly includes the following parts: (1) model pool evolution; (2) dynamic model assembling; (3) model pool update strategies.
>     - For (1), we proposed a NOVEL model evolution approach through linking likelihood under Bayesian theory to fitness value under evolutionary computational theory. It is significantly different from DyEnsemble, which only has a fixed model pool and its models are generated through neuron dropout and weight perturbation, which only focus on the problem of noise.
>     - For (2), we mainly employed the framework of DyEnsemble. By introducing an evolvable model pool, we further extend the framework to solve the problem of functional changes, which can not be well addressed by DyEnsemble (As shown in Fig.2).
>     - For (3), we proposed two NEW strategies, namely evolve-at-changes and history-model-archive, to balance the trade-off between efficiency and accuracy as well as improve the stability of state estimation. These operations are very critical for the effectiveness of our approach.
>
> 2. ***“The DE algorithm tends to evolve all models towards the same one, is this evolution approach really better than a well-defined model pool, considering we may can utilize the statistical features of neural signals to well define a good model pool?”***
> - Indeed, the weakness of DE lies in that the models evolve to the same one. Thus, we employ the idea of JADE where the historical models are archived so that the pool can maintain the diversity. Experiments demonstrated the effectiveness of history-model-archive strategy. Besides, the EvoEnsemble aim to provide a new framework to link evolutionary computing with dynamic Bayesian estimation, to solve the problem of functional changes. Using DE as the evolutionary model is an implementation of EvoEnsemble, and different evolutionary models can be applied in the framework.
> - It is difficult to construct a well-defined model pool, given that the functions in the test stage can get out of the range of functions in the training stage, i.e., the function in the test stage may not be covered by the training data. In this situation, using the features of neural signals in the training data is insufficient to build the model pool that generates to the incoming data. Thus, instead of relying on a predefined model pool, EvoEnsemble evolves the models according to the incoming neural signals if needed. In this way, EvoEnsemble tracks functions even if they are out of training data.
>
> 3. ***“How about we restrict models in the pool to be disentangled (such as orthogonal base functions), will this works better?”***
> - This can be a very interesting problem for further discussion. Restricting models to be disentangled may have good decoding performance if we use other ensemble methods such as stacking. However, we assemble the models with Bayesian model averaging rule, which requires the correct data generating model to be on the list of models under consideration [[Clarke, 2003](https://www.jmlr.org/papers/volume4/clarke03a/clarke03a.pdf)]. That is, our model pool needs to be ‘good but different’, and there should be at least a few models with high likelihood. In future study, we may try to use the orthogonal base functions to see how it works. Many thanks.

---

> > ### Author Response · Authors · 2022-08-02
> > **Additional Response to Reviewer Lhh8**
> >
> > 4. ***"How to understand Fig. 2 condition 3, the $h_t^2$ tracking result is quite different from the ground truth, although the likelihood and $\text{R}^2$ are high?"***
> > - This is because the ground truth of condition 3 is slow non-monotonic change. That is, the $h_t^2$ do not change sharply; hence the function parameters estimated by EvoEnsemble are enough to handle the decoding problem well. And so does condition 1, whose ground truth is slow monotonic change. Note that, the y-scale between the five conditions is different.
> >
> > 5. ***"The computational efficiency of evolutionary computation."***
> > - The computational costs of one model evolution generation process are $1.130 \pm 	0.172$ s, $0.489 \pm 	0.109$ s, $0.519 \pm 0.122$ s, $0.509 \pm 	0.120$ s, for Data-M, Data-P1, Data-P2 and Data-P3 respectively. With the evolve-at-changes strategy, model evolution is only perform-as-needed, which largely decrease the computational costs. Parallel computing-based implementation can further speed up the computation. Specifically, we can separate the model evolution and the dynamic model assembling in two processes, where the model pool update module runs silently in the background, collecting data and evolving models. Once it finishes updating, replace the original model pool. Such parallel operations can enable online real-time decoding.
> >
> > 6. ***"In neural decoding tasks, the trend of state x is essential for the application. Still, the decoding results, which always seem to fluctuate up and down with respect to the ground truth, influence the decoding effect."***
> > - In the manuscript, we show the tracking performance of the functional changes $h_t(\cdot)$ rather than the decoding state $x_t$. It is true that the trend of state $x$ is essential for the application, but this is guaranteed by the state transition function $g(\cdot)$ rather than the observation function $h_t(\cdot)$.

---

### Official Review · Reviewer_YBUd · 2022-07-11

**Rating:** 6
**Confidence:** 2
**Soundness:** 3 good
**Presentation:** 3 good
**Contribution:** 3 good

**Summary:**

The work addresses the task of neural decoding (estimating intended user action from observed neural brain signals) for a changing observation model ($h_t$ in eq. 1) over time. The method involves initially training an ensemble model via minimizing a squared error loss on training data (section 3.1.1) and then dynamically evolving models in the pool towards members with a high observation likelihood score (fitness measure) computed from the current observed neural signal (sections 3.1.2-3.1.3). This adaptive procedure is iterated over time. The overall model prediction is then made from a weighted combination of ensemble members by Bayesian model averaging (section 3.2).

Using this learned and dynamically evolving observation model, the $x_t$ state (ex: intended cursor velocity) is estimated using a particle filter (section 3.3).

Finally, two additional strategies are employed to control the model pool update frequency and encourage diversity (section 3.4). The __evolve-at-changes__ strategy only updates the model pool once a running average of the likelihood drops below a certain threshold. The __history-model-archive__ strategy enforces that a certain proportion of historic models be kept in the model to encourage diversity.

An ablation study of the different model components is done in experimental section 4.1 involving an artificially generated observation model.

Evaluation on real neural signal datasets and against other adaptive neural decoder schemes (Kalman filter with fast recalibration [13] and DyEnsemble [17]) is done in section 4.2. The proposed EvoEnsemble method yields improved performance compared to baselines.


**Questions:**

__This question is copied from the Weaknesses section:__
I feel it would be useful to have more intuition as to why this approach beats the Kalman filter with recalibration method [13]. In the introductory section, it is stated that this method “usually can not cope with changes in the short-term such as single trials” although this is more an assert of an end-effect (poor performance) instead of a reasoning for it. Discussion in the experimental section likewise does not seem to go into depth as to why EvoEnsemble outperforms this baseline (bottom of page 8). It is stated that the Kalman filter with recalibration is set to update every 20 slots. Was this value optimally chosen and is it similar to the rate that EvoEnsemble determines using its dynamic update strategy? Put another way, is EvoEnsemble beating out this method due to difference in update rate or does is update mechanism itself responsible for the delta in performance? Is there further intuition as to why EvoEnsemble outperforms this baseline.


**Limitations:**

The authors briefly state in the Conclusion section that “One limitation of our approach is the computational efficiency of the evolution process, since it requires hundreds of generations for one model pool update process. Our future work may focus on improving the efficiency of evolutionary computation.”

Likewise - although not a major limitation – I would question how much manual hyperparameter tuning may be required to achieve optimal performance per dataset. Ideally these could be limited.


**Strengths And Weaknesses:**

__General Strengths:__
- The dynamic ensemble scheme successfully updates to changes in the neural observation model. Furthermore – from my understanding – updates are done based solely on the observed neural signal whereas other methods such as the Kalman filter with fast recalibration uses a $z_t$ surrogate (indicating the direction to target) as a surrogate for the hidden $x_t$ state to do recalibration.
- I found the experimental ablation study (section 4.1) particularly insightful in illustrating the effects of the different method additions.
- Beats other adaptive neural decoder baselines.

__General Weaknesses:__
- There seems to be a relatively larger number of steps and hyperparameters which are empirically motivated and I would potentially be concerned how much manual hyperparameter tuning may be required to achieve optimal performance per dataset.  This seems especially true for the strategies for model pool updating (section 3.4).
- I feel it would be useful to have more intuition as to why this approach beats the Kalman filter with recalibration method [13]. In the introductory section, it is stated that this method “usually can not cope with changes in the short-term such as single trials” although this is more an assert of an end-effect (poor performance) instead of a reasoning for it. Discussion in the experimental section likewise does not seem to go into depth as to why EvoEnsemble outperforms this baseline (bottom of page 8). It is stated that the Kalman filter with recalibration is set to update every 20 slots. Was this value optimally chosen and is it similar to the rate that EvoEnsemble determines using its dynamic update strategy? Put another way, is EvoEnsemble beating out this method due to difference in update rate or does is update mechanism itself responsible for the delta in performance? Is there further intuition as to why EvoEnsemble outperforms this baseline.

__Originality and Significance:__
To the best of my knowledge, the work is an original novel contribution to using an evolutionary ensemble for neural decoder and beats similar baselines.

__Quality:__
The quality is good. The ablation study on simulated data (section 4.1) highlights the application of the different method components. This is followed by practical evaluation against related baseline methods on several real neural datasets (section 4.2).

__Clarity:__
In general, the submission is clear. There are a few sections with somewhat awkward wording though this is minor.

__Minor Additional Feedback:__
At the top of page 3 (lines 83-84), the text seems to indicate that Fig. 1 is simultaneously above and below.

---

> ### Author Response · Authors · 2022-08-02
> **Response to Reviewer YBUd**
>
> Many thanks for the generally positive comments and feedback. More experiments are carried out to further compare with KF (recalib) and more explanation is added to explain why our approach beat KF (recalib). Hopefully these will resolve most of your remaining concerns.
>
> ***“I feel it would be useful to have more intuition as to why this approach beats the Kalman filter with recalibration method [[Brandman et al.,2018](https://iopscience.iop.org/article/10.1088/1741-2552/aa9ee7/pdf?casa_token=dDIJQaHj6vsAAAAA:wdwkIREfdqrro2LwAQg-V4V6sCUSKwzSFjyytWuO6yn_DU_ERlTl4k_8Hq7dUk46i4dhMy8LdUGqVCEJFLU)]. It is stated that the Kalman filter with recalibration is set to update every 20 slots. Was this value optimally chosen and is it similar to the rate that EvoEnsemble determines using its dynamic update strategy? Put another way, is EvoEnsemble beating out this method due to difference in update rate or does is update mechanism itself responsible for the delta in performance? Is there further intuition as to why EvoEnsemble outperforms this baseline.”***
>
> - Experiments are carried out to compare EvoEnsemble with KF (recalib) with different update timings. In Table R2, KF (recalib-X) represents updating the decoder every X time slots, and KF (recalib-adapt) represents the update time of KF keeping inline with the EvoEnsemble. Overall, KF (recalib) with small updating slots (below 10) achieves the best performance. However, since the recalibration of KF requires the true trajectory for training, it is usually impractical to use such a small updating slot in online BCI systems.
> - Indeed, intuitively KF (recalib) should perform better because it uses the true trajectory at each recalibration step while EvoEnsemble does not. However, the recalibration of KF (recalib) uses all the historical data. With the situation that the function changes in time, KF (recalib) can only learn a mixture of functions in different time slots, thus can not be accurate at each time slot. On the contrary, EvoEnsemble learns the accurate functions at different time slots to build the model pool, and selects the optimal model for prediction, which is theoretically more accurate than using a single mixture model. The evolution process further improves the performance when the function changes out of the model pool. Thus, EvoEnsemble outperforms KF (recalib) even if it does not use true trajectory for updating.
>
> **Table R2: Decoding performance (CC) of EvoEnsemble and KF(recalib) with different update duration.**
> | Decoder           | Data-M | Data-P1 | Data-P2 | Data-P3 |
> |-------------------|--------|---------|---------|---------|
> | KF(recalib-1)     | **0.734**  | **0.627**   | **0.627**   | **0.613**   |
> | KF(recalib-5)     | 0.706  | 0.571   | 0.602   | 0.564   |
> | KF(recalib-10)    | 0.707  | 0.525   | 0.593   | 0.520   |
> | KF(recalib-20)    | 0.705  | 0.477   | 0.548   | 0.439   |
> | KF(recalib-50)    | 0.711  | 0.443   | 0.516   | 0.400   |
> | KF(recalib-100)   | 0.713  | 0.428   | 0.511   | 0.379   |
> | KF(recalib-adapt) | 0.701  | 0.495   | 0.559   | 0.473   |
> | EvoEnsemble       | **0.764**  | **0.547**   | **0.561**   | **0.487**   |

---

### Official Review · Reviewer_g44o · 2022-07-15

**Rating:** 5
**Confidence:** 2
**Soundness:** 3 good
**Presentation:** 4 excellent
**Contribution:** 3 good

**Summary:**

The authors propose an evolutionary ensemble method for decoding neural signals. The key challenge that this work addresses is the fact that the underlying function between the variable of interest and the signal changes over time ("functional changes"). The authors suggest that their method can discover and adapt to functional changes and tested its performance on synthetic and neural datasets.

**Questions:**

see above.

**Limitations:**

see above.

**Strengths And Weaknesses:**

Strength: the study is well motivated, and the algorithm proposed is well defined and well analyzed.

Weakness: I would like to see performance comparison against more modern ML methods. I am no expert in the area but I would surprised if OLE and Kalman filters are meaningful targets for comparison. The following review (https://www.ncbi.nlm.nih.gov/pmc/articles/PMC7470933/pdf/ENEURO.0506-19.2020.pdf) appears to suggest that there are many NN based algorithms that take the functional change problem into account.

---

> ### Author Response · Authors · 2022-08-02
> **Response to Reviewer g44o**
>
> Many thanks for the helpful comments and feedback. We have carried out extra experiments to compare the performance of EvoEnsemble with more machine learning methods. Hopefully, these will address most of your concerns, and they can be taken into consideration when deciding the final score of the paper.
>
> ***"I would like to see performance comparison against more modern ML methods. I am no expert in the area but I would surprised if OLE and Kalman filters are meaningful targets for comparison. The following review appears to suggest that there are many NN based algorithms that take the functional change problem into account."***
>
> - As suggested, experiments are carried out to compare the methods mentioned in the review [[Glaser et al., 2020](https://www.eneuro.org/content/eneuro/early/2020/07/31/ENEURO.0506-19.2020.full.pdf)] (see Table R1). The results demonstrate that EvoEnsemble outperforms the NN models, including ANN, RNN, GRU and LSTM. Although nonlinear decoders such as GRU and LSTM usually contain higher learning ability, they assume that the training and test data are from the same distribution, and the model is fixed after training. Although LSTM shows dynamic properties with the gating mechanisms, it only learns functions that are presented in the training data. Given the conditions that the functions change out of the range of the training data, the performance of GRU and LSTM decays. While with EvoEnsemble, the model adjustment is directly driven by the changes in data, so that it can track functional changes out of the training data.
>
> **Table R1: Decoding performance (CC) of different decoders.**
> | Decoder   | Condition 1       | Condition 2       | Condition 3       | Condition 4       | Condition 5       |
> |-----------|-------------------|-------------------|-------------------|-------------------|-------------------|
> | WF        | 0.795 $\pm$ 0.040 | 0.441 $\pm$ 0.002 | 0.930 $\pm$ 0.020 | 0.352 $\pm$ 0.001 | 0.952 $\pm$ 0.000 |
> | WC        | 0.795 $\pm$ 0.040 | 0.441 $\pm$ 0.003 | 0.930 $\pm$ 0.020 | 0.352 $\pm$ 0.009 | 0.951 $\pm$ 0.000 |
> | SVR       | 0.853 $\pm$ 0.023 | 0.031 $\pm$ 0.010 | 0.922 $\pm$ 0.007 | 0.369 $\pm$ 0.005 | 0.890 $\pm$ 0.000 |
> | ANN       | 0.878 $\pm$ 0.113 | 0.420 $\pm$ 0.010 | 0.870 $\pm$ 0.023 | 0.348 $\pm$ 0.006 | 0.835 $\pm$ 0.140 |
> | RNN       | 0.836 $\pm$ 0.091 | 0.437 $\pm$ 0.010 | 0.865 $\pm$ 0.068 | 0.350 $\pm$ 0.005 | 0.910 $\pm$ 0.095 |
> | GRU       | 0.697 $\pm$ 0.057 | 0.428 $\pm$ 0.006 | 0.792 $\pm$ 0.043 | 0.354 $\pm$ 0.001 | 0.953 $\pm$ 0.022 |
> | LSTM      | 0.843 $\pm$ 0.039 | 0.431 $\pm$ 0.007 | 0.947 $\pm$ 0.005 | 0.335 $\pm$ 0.002 | 0.856 $\pm$ 0.100 |
> | Evo(ours) | **0.894 $\pm$ 0.007** | **0.905 $\pm$ 0.004** | **0.952 $\pm$ 0.037** | **0.895 $\pm$ 0.004** | **0.997 $\pm$ 0.001** |

---

### Meta-Review · Area_Chair_3CTK · 2022-08-26

**Recommendation:** Accept
**Confidence:** Less certain

**Metareview:**

The review ratings/confidences were 5/2, 6/2, 4/4, and 5/2. Although the average rating of 5 was just above the acceptance threshold, I think that it should somehow be discounted by the lower confidence levels. Although I myself does not have expertise in the field of BCI, as for the reviewers' evaluation, I think that they basically agreed on the following points:
- The problem is well motivated.
- The proposed method was built on DyEnsemble with some empirically-motivated extensions to have decoder models dynamically evolving. One could then argue that the proposal is not groundbreaking but somehow incremental.
- The authors showed experimentally that the proposed method works well compared with a number of other existing methods.

I also noticed that the authors made revision (adding a paragraph at the end of Section 2: It can be observed in the August 10 revision), which would have improved readability of this paper. I would thus recommend acceptance of this paper, provided that there is room for it.

**Award:**

No

---

### Decision · Program_Chairs · 2022-09-14

Accept